# FedPnP: A Plug and Play Approach For Personalized Graph-Structured Federated Learning

## Abstract

In Personalized Federated Learning (PFL), existing methods often overlook the intricate interconnections between clients and their local datasets, limiting effective information sharing. In this work, we introduce "FedPnP", a novel approach that leverages the inherent graph-based relationships among clients. Clients connected by a graph tend to exhibit similar model responses to similar input data, leading to a graph-based optimization problem linked to inverse problems like compressed sensing. To tackle this optimization problem, we employ a Half-Quadratic-Splitting technique (HQS) to effectively decompose it into two subproblems. The first subproblem, acting as a data fidelity term, ensures local models perform well on their respective datasets, while the second, serving as a sparsity-inducing term, promotes the smoothness of local model weights on the graph. Notably, we introduce a *structural proximal term*, a generalization for FedProx, in the first subproblem and demonstrate that any graph denoiser with a controllable noise parameter can be integrated as the second subproblem, offering flexibility without explicit derivation. We evaluate FedPnP on computer vision datasets (CIFAR-10, MNIST) and a human activity recognition dataset (HAR-BOX) to test its performance in real-world PFL scenarios. Empirical results confirm that FedPnP outperforms state-of-the-art algorithms. This novel bridge between PFL and inverse problems opens up the potential for cross-pollination of solutions, yielding superior algorithms for PFL tasks.

## 1 Introduction

Distributed machine learning has received significant attention and extensive research in recent years (Liu et al., 2022a; Wen et al., 2023). The concept of collaborative learning has become increasingly appealing due to the proliferation of high-performance machinery and advanced computing infrastructure in the digital era (Liu et al., 2022c; Huang et al., 2022b). This growth is fueled by remarkable advancements in personal electronic and hand-held devices, including smartphones, wearable devices, home assistants, and autonomous vehicles (Jones et al., 2020). These interconnected devices are equipped with sensors to collect vast amounts of data and employ various technical means to enable automatic, intelligent computing.

One widely adopted approach in distributed machine learning is Federated Learning (FL), introduced by McMahan et al. (McMahan et al., 2017). FL has garnered substantial interest and application across various domains, including smart cities (Pandya et al., 2023), computer vision (Chen & Chao, 2021; Oh et al., 2021; Liu et al., 2022b; Xiong et al., 2023), healthcare (Antunes et al., 2022), and finance (Wu et al., 2022). Its ability to leverage distributed data without centralization makes FL an attractive solution for addressing challenges related to privacy, security, and scalability in machine learning. FL collaboratively trains local models owned by different participants, preserving privacy and reducing communication (McMahan et al., 2017). Conventional FL aims to minimize collective loss through global model training, but non-i.i.d. statistical diversity among clients can impede its suitability (Gao et al., 2022). Conversely, training without FL results in a loss of local model generalization due to inadequate data (T Dinh et al., 2020). To address this challenge, various techniques have emerged to strike a balance between globalization and model personalization within the realm of Personalized Federated Learning (PFL) (T Dinh et al., 2020; Ozkara et al., 2022; Setayesh et al., 2022; Ma et al., 2022b). The primary objective of PFL is to train local models that excel on local datasets while maintaining adequate generalization. This is typically achieved through

methods such as fine-tuning the global model (Zhang et al., 2022), introducing a regularization term to the objective function (regularized PFL) (T Dinh et al., 2020), clustering participants (clustered PFL) (Zhang et al., 2020), decoupling parameters to tailor customized models for participants, utilizing knowledge distillation (architecture-based PFL) (Wang et al., 2020; Ma et al., 2022a; Huang et al., 2022a), and employing Meta-learning PFL (Yang et al., 2023).

Our framework focuses on regularized PFL. Existing methods like FedProx (Li et al., 2020a) fall short of capturing fine-grained participant relationships , e.g., relationships between participants $i$ and $j$ and the connection strength. The sole structured regularization method (Chen et al., 2022) applies constraints to neighboring participants based on the graph but overlooks indirect connections and the complete graph. Additionally, cluster-based PFL, despite creating group-wise graphs, lacks insight into the structured interactions among individual participants, primarily capturing coarse relationships (Ouyang et al., 2022).

To address the limitations of regularized and cluster-based PFL approaches, we introduce a participant relationship-based graph construction. This relationship can encompass various factors such as dataset similarity, hardware connections, or social interactions among participants. We contend that participants linked by this graph should exhibit similar behavior in their local models, reflecting the underlying structure. We focus on creating such a relationship based on dataset similarity. Therefore, participants sharing a dataset similarity connection imply similar data distributions, resulting in analogous responses to comparable input samples in their local models. This observation leads us to conjecture that the model weights of graph-connected participants exhibit sparsity in some domain—a notion equivalent to the "smoothness of the graph" imposed by a graph denoiser or the idea that the "graph signal" (i.e., model weights) should exhibit smoothness across the graph. In this scenario, the local loss function corresponds to data fidelity, while graph smoothness represents the sparsity of the graph signal, i.e., local model weights. This insight reveals an unexplored connection between our approach and well-established concepts in inverse problems, as seen in Compressed Sensing (CS) (Donoho, 2006) and imaging domains (Song et al., 2022). In inverse problems, the primary objective is to recover an unknown signal, image, or multi-dimensional volume from observations that have undergone an ill-posed forward process, i.e., forward model. Consequently, arriving at a unique reconstruction is often unattainable. To better approximate the unknown data, prior knowledge in the form of a sparsity-inducing term is added to the optimization problem, typically manifesting as smoothness on the signal (Ongie et al., 2020). For instance, medical MRI images often exhibit smoothness properties in domains like the discrete Fourier transform, as highlighted by Zhong et al. (Zhong et al., 2022) .

In this work, we propose to address graph-structured PFL as an inverse problem. We argue that PFL can be likened to solving a general inverse problem where the goal is to recover model weights that adhere to both the forward model (local loss function) and exhibit sparsity in a specific domain (graph smoothness). Moreover, the proposed approach generalizes both the renowned FedAvg and FedProx algorithms. Consequently, this paper aims to answer a pivotal question: *Can enhancing model personalization be achieved through structured collaborative optimization, leveraging solutions from well-established inverse problems?*

The contribution of our work is as follows:

**[C1] Graph-Structured Federated PnP.** We propose a novel approach called Graph-structured Federated Plug & Play (FEDPNP) to solve the PFL and data heterogeneity problems. The proposed FEDPNP introduces a *structured proximal term* serving as regularization, addressing statistical heterogeneity in the FL framework based on participants' graph-based interconnections. Our algorithm further generalizes both renowned FedAvg (McMahan et al., 2017) and FedProx (Li et al., 2020a).

**[C2] A New Perspective.** We provide a fresh perspective on PFL algorithms, examining local and global updates structurally within the context of inverse problem solutions. FEDPNP is a culmination of the PnP Half-Quadratic-Splitting (HQS) algorithm, comprising two separate sub-problems where a graph denoiser can be readily *Plugged in* as the second optimization solution. To the best of our knowledge, this is the first work connecting PFL and inverse problems. Such a link paves the way for the mutual exchange of solutions, possibly leading to the development of superior PFL algorithms.

**[C3] Experimental Evaluation.** We have conducted experiments on computer vision datasets like MNIST and CIFAR-10, along with a real-world human activity recognition dataset, HARBOX, that captures the data heterogeneity in practical cases, making it suitable for evaluating PFL performance.

## 2 METHODOLOGY

In this section, we formalize the environment specification and the problem description, with a specific focus on an FL solution involving multiple participants collaboratively training ML models.

### 2.1 ENVIRONMENT DESCRIPTION

In an FL system with $K$ participants or clients, each possesses a local dataset denoted as $D_k$, which is drawn from distinct distributions $P_k$ and is typically non-identically and independently distributed (non-IID). To capture the interconnections among participants, we define a graph $\mathcal{G}$ with an adjacency matrix $\boldsymbol{A}$, reflecting the topological relationships within the FL system. This adjacency matrix $\boldsymbol{A}$ represents the connections among participants and is constructed based on the similarity of $D_k$. More details on graph construction can be found in the supplementary section A.5.

### 2.2 PROPOSED FRAMEWORK

Generally speaking, the FL attempts to optimize global model weights in each communication round. Let $\boldsymbol{\omega}_k \in \mathbb{R}^B$ be the model weight vector for the $k^{\text{th}}$ client, where $B$ is the number of parameters for the ML model. The general objective of the FL is

$$\min_{\boldsymbol{\omega}_{1:K}} F\left(f_1(\boldsymbol{\omega}_1), ..., f_K(\boldsymbol{\omega}_K)\right), \tag{1}$$

where $F(\cdot)$ is the aggregator function for the local loss functions $f_k(\boldsymbol{\omega}_k), k = 1, ..., K$. In Conventional FL (McMahan et al., 2017), the aggregator function is considered to be the weighted summation of local objectives, i.e., $F(\{f_k(\boldsymbol{\omega}_k)\}_{k=1}^K) = \sum_{k=1}^K \zeta_k f_k(\boldsymbol{\omega}_k)$ with $\zeta_k = \frac{|D_k|}{\sum_{k=1}^K |D_k|}$. However, such an objective function proves to miss various characterizations of the local datasets, such as statistical heterogeneity and local model personalization.

To address the limitations of the objective function in equation 1, personalized optimization can be framed as a bi-level optimization. Thus, for each client, the optimum model weights are derived as

$$\begin{aligned}
\min_{\boldsymbol{\omega}_{1:K}} \quad & f_k(\boldsymbol{\omega}_k) + r_k(\boldsymbol{\omega}_k, \boldsymbol{\omega}_{\mathrm{G}}) \\
\text{s.t.} \quad & \boldsymbol{\omega}_{\mathrm{G}} = \arg\min_{\boldsymbol{\omega}} F\left(f_1(\boldsymbol{\omega}), ..., f_K(\boldsymbol{\omega})\right)
\end{aligned} \tag{2}$$

where $r_k(\boldsymbol{\omega}_k, \boldsymbol{\omega}_{\mathrm{G}})$ represents the regularization term applied to constrain the impact of local updates and help with model personalization. In addition, $\boldsymbol{\omega}_{\mathrm{G}}$ is the global model weight obtainable using conventional FL algorithms such as FedAvg (McMahan et al., 2017). However, this optimization approach lacks structured information about participant interconnections and only maintains the personalized model close to the global model.

#### 2.2.1 PERSONALIZED FL AS AN INVERSE PROBLEM

In PFL, optimization involves two crucial aspects: personalization (local updates) and knowledge sharing (global updates). Personalization happens as models are updated on local datasets, while knowledge sharing involves aggregating, replicating, and distributing updated local models among participants. These aspects can be viewed from different perspectives.

Let $\mathbb{W}_k$ be the subspace of model weights minimizing $f_k(\boldsymbol{\omega}_k)$ with sufficient accuracy. Local training, using methods like stochastic gradient descent, guides $\boldsymbol{\omega}_k$ toward $\mathbb{W}_k$, behaving like a fidelity term. Knowledge sharing, conversely, moves $\boldsymbol{\omega}_k$ away from $\mathbb{W}_k$ but ensures smoothness among collective model weights; e.g., FedAvg (McMahan et al., 2017) makes all $\boldsymbol{\omega}_k$ in $W = [\boldsymbol{\omega}_1, ..., \boldsymbol{\omega}_K]$ identical after aggregation, ensuring sparsity. This resembles sparsity-inducing terms because smoothing renders vector $W$ sparse. Similar notions arise in inverse problems, with fidelity terms aligning the reconstructed signal with the model subspace and regularization terms ensuring signal smoothness. This shows a close connection between PFL objectives and general inverse problem formulations.

Furthermore, the graph $\mathcal{G}$ captures the structural relationships among participants, suggesting that connected local models should behave similarly based on this structure. For instance, participants structured by data similarity indicate similar data distributions, leading to similar responses from their local models to comparable inputs. Consequently, we hypothesize that such model weights exhibit smoothness on the graph (Liu et al., 2023). Utilizing this prior knowledge ensures personalized local models while structurally updating the optimization problem.

Assume that $W[i] = [\boldsymbol{\omega}_1[i], ... \boldsymbol{\omega}_K[i]] \in \mathbb{R}^K, \forall i = 1, ..., B$ represents the graph signal weight vector, where $\boldsymbol{\omega}_k[i]$ signifies the $i^{\text{th}}$ model weight of the $k^{\text{th}}$ participant. Define $s(W[i])$ as the graph-structured sparsity-promoting function, e.g., Total Variation (TV) regularization, $\text{TV}(W[i])$. Under such a prior term, the objective function is determined as:

$$\mathcal{P}1: \widehat{W}[i] = \arg\min_{W[i]} F\left(f_1(\boldsymbol{\omega}_1), ..., f_K(\boldsymbol{\omega}_K)\right) \quad s.t. \quad s(W[i]) \leq \epsilon' \quad i = 1, ... B, \tag{3}$$

where $\epsilon' \geq 0$. Upon closer examination, $(\mathcal{P}1)$ shares similarities with inverse problems (Bouman & Buzzard, 2023). In essence, we aim to project (or approximate) $W[i]$ onto the weight subspace, minimizing the first term, $F(\cdot)$, while ensuring sufficient sparsity as required by the second term. This connection with inverse problems offers opportunities to adapt various algorithms and techniques for learning and recovering model weights. We term this problem the *Distributed Problem*. Involving the constraint into the optimization using the lagrangian multiplier $\beta$, we present the optimization problem as

$$\mathcal{P}1: \widehat{W}[i] = \arg\min_{W[i]} F\left(f_1(\boldsymbol{\omega}_1), ..., f_K(\boldsymbol{\omega}_K)\right) + \beta s(W[i]) \quad i = 1, ... B, \tag{4}$$

where $\beta$ has the equivalence role as $\epsilon$ and imposes the degree of sparsity in the objective function. To simplify the optimization, we choose the two terms for the prior information as $s(\boldsymbol{\Omega}) = L_{\text{smooth}} + L_{\text{linear}}$ specified as follows:

$$\mathcal{P}1: \widehat{\boldsymbol{\Omega}} = \arg\min_{\boldsymbol{\Omega}} F\left(\mathbb{F}(\boldsymbol{\Omega})\right) + \widehat{\beta}_1 \boldsymbol{\Omega}^T \boldsymbol{L} \boldsymbol{\Omega} + \widehat{\beta}_2 ||\boldsymbol{\Omega}^T \boldsymbol{L}||_2^2, \tag{5}$$

where $\boldsymbol{\Omega} = [W[1], ..., W[B]] \in \mathbb{R}^{K \times B}$ is the weight matrix with the rows corresponding to the weights of the $k^{\text{th}}$ participant, and $\mathbb{F}(\boldsymbol{\Omega}) = \{f_k(\boldsymbol{\omega}_k)\}_{k=1}^K$ is the set of local loss functions. Moreover, $\boldsymbol{L}$ is the graph Laplacian of $\boldsymbol{A}$ defined as $\boldsymbol{L} = \boldsymbol{D} - \boldsymbol{A}$ where $\boldsymbol{D}$ represents the diagonal degree matrix of $\boldsymbol{A}$. The objective term $L_{\text{smooth}} = \boldsymbol{\Omega}^T \boldsymbol{L} \boldsymbol{\Omega}$ is incorporated to enforce sparsity in the form of smoothness of the graph signal. Additionally, $L_{\text{linear}} = ||\boldsymbol{\Omega}^T \boldsymbol{L}||_2^2$ is introduced to ensure minimal deviation of $W[i]$ from the linear form, as referenced in (Stanković et al., 2019).

### 2.2.2 PLUG & PLAY FL (FEDPNP): A SOLUTION TO GRAPH-STRUCTURED PFL AS AN INVERSE PROBLEM

The *distributed problem* in $(\mathcal{P}1)$ seeks to minimize local losses while considering graph signal smoothness, where the aggregator $F(\cdot)$ is pivotal. Notably, the graph signal corresponds to the participants' model weights. However, solving $(\mathcal{P}1)$ directly proves challenging and impractical in real-world scenarios due to two primary factors. First, the aggregator function $F\left(\mathbb{F}(\boldsymbol{\Omega})\right)$ operates in a one-dimensional space, while both $L_{\text{smooth}}$ and $L_{\text{linear}}$ have $B$ dimensions, causing a dimensionality mismatch. Secondly, a portion of the objective of $F\left(\mathbb{F}(\boldsymbol{\Omega})\right)$ can only be optimized locally, while both $L_{\text{smooth}}$ and $L_{\text{linear}}$ necessitate the availability of all local weights. Consequently, there exists a disparity in the optimization requirements between these components.

To circumvent the challenges associated with directly solving the minimization problem $(\mathcal{P}1)$, we propose an alternative approach that involves decoupling the problem using the Half-Quadratic-Splitting (HQS) technique. HQS has garnered substantial attention and has been extensively employed in prominent domains, including Compressed Sensing (CS) (Rasti-Meymandi et al., 2023), image restoration (Zhang et al., 2021), and sparse recovery (Dong et al., 2018). HQS is solving the following minimization problem by adding an augmented variable

$$\mathcal{P}2: \widehat{\boldsymbol{\Psi}}, \widehat{\boldsymbol{\Omega}} = \arg\min_{\boldsymbol{\Omega}, \boldsymbol{\Psi}} F\left(\mathbb{F}(\boldsymbol{\Omega})\right) + \widehat{\beta}_1 \boldsymbol{\Psi}^T \boldsymbol{L} \boldsymbol{\Psi} + \widehat{\beta}_2 ||\boldsymbol{\Psi}^T \boldsymbol{L}||_2^2 \quad s.t \quad \boldsymbol{\Psi} = \boldsymbol{\Omega}, \tag{6}$$

where $\boldsymbol{\Psi} \in \mathbb{R}^B$ denotes the auxiliary vector variable. Using the penalty method we have

$$\mathcal{P}3: \widehat{\boldsymbol{\Psi}}, \widehat{\boldsymbol{\Omega}} = \arg\min_{\boldsymbol{\Omega}, \boldsymbol{\Psi}} F\left(\mathbb{F}(\boldsymbol{\Omega})\right) + \widehat{\beta}_1 \boldsymbol{\Psi}^T \boldsymbol{L} \boldsymbol{\Psi} + \widehat{\beta}_2 ||\boldsymbol{\Psi}^T \boldsymbol{L}||_2^2 + \frac{\mu}{2} ||\boldsymbol{\Psi} - \boldsymbol{\Omega}||_2^2. \tag{7}$$

The optimization problem can further be expressed in disjoint blocks as

$$\mathcal{P}4: \begin{cases} \widehat{\boldsymbol{\Omega}}^t = \arg\min_{\boldsymbol{\Omega}} F\left(\mathbb{F}(\boldsymbol{\Omega})\right) + \frac{\mu}{2} ||\widehat{\boldsymbol{\Psi}}^{t-1} - \boldsymbol{\Omega}||_2^2 \\ \widehat{\boldsymbol{\Psi}}^t = \arg\min_{\boldsymbol{\Psi}} \widehat{\beta}_1 \boldsymbol{\Psi}^T \boldsymbol{L} \boldsymbol{\Psi} + \widehat{\beta}_2 ||\boldsymbol{\Psi}^T \boldsymbol{L}||_2^2 + \frac{\mu}{2} ||\boldsymbol{\Psi} - \widehat{\boldsymbol{\Omega}}^t||_2^2 \end{cases}, \tag{8}$$

where the superscript $t$ represents the current communication round. Therefore, solving the optimization problem in $(\mathcal{P}3)$ is now achievable by iterating through the first and second terms in $(\mathcal{P}4)$.

The first expression in $(\mathcal{P}4)$ represents the general objective function in equation 1, enriched with an intriguing proximal term. The term $\frac{\mu}{2}||\widehat{\boldsymbol{\Psi}}^{t-1}-\boldsymbol{\Omega}||_2^2$ implies that local model weight optimization is regularized to stay close to structurally tuned model weights. This concept aligns with the FedProx algorithm introduced in (Li et al., 2020a), which uses a proximal term to mitigate dataset heterogeneity and partial information by considering the proximity of local weights to global weights. However, our proximal term differs significantly in two ways: firstly, $\widehat{\boldsymbol{\Psi}}^{t-1}$ relates to the second objective term in $(\mathcal{P}4)$, and secondly, $\widehat{\boldsymbol{\Psi}}^{t-1}$ is updated in a structural, not global, manner. In essence, we introduce a *structured proximal term* into the objective function, which partici-

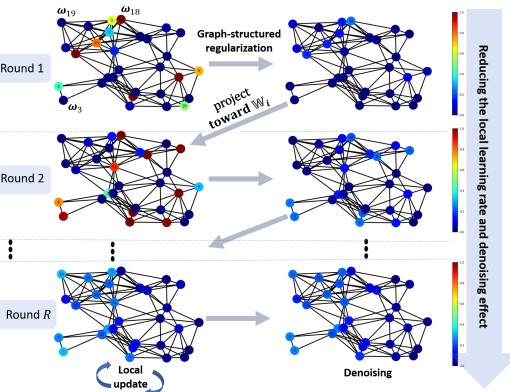

Figure 1: An illustration of the mechanics behind the proposed FEDPNP. Each node is a model weight connected via a graph.

pants adhere to during local updates. Thus, the $k^{\text{th}}$ participant minimizes their local objective using a solver of their choice, constrained by the structurally acquired model weights as

$$\min_{\boldsymbol{\omega}_k} l_k(\boldsymbol{\omega}_k^t; \widehat{\boldsymbol{\Psi}}^{t-1}[k]) = f_k(\boldsymbol{\omega}_k^t) + \frac{\mu}{2}||\widehat{\boldsymbol{\Psi}}^{t-1}[k] - \boldsymbol{\omega}_k^t||_2^2, \tag{9}$$

where $\widehat{\boldsymbol{\Psi}}^{t-1}[k] \in \mathbb{R}^B$ is the $k^{\text{th}}$ row of $\widehat{\boldsymbol{\Psi}}^{t-1}$ solved from the second subproblem in $(\mathcal{P}4)$, i.e., the structural model weight (as opposed to global model weight) tailored specifically for participant $k$ according to the graph's topology.

The second term in $(\mathcal{P}4)$ is a convex problem. The closed-form solution to this problem is derived by taking the gradient of the objective function as follows:

$$\nabla_{\boldsymbol{\Psi}} \widehat{\boldsymbol{\Psi}}^t = \mu(\boldsymbol{\Psi} - \widehat{\boldsymbol{\Omega}}^t) + 2\widehat{\beta}_1 \boldsymbol{L}\boldsymbol{\Psi} + 2\widehat{\beta}_2 \boldsymbol{L}^2\boldsymbol{\Psi} = 0 \Longrightarrow \tag{10}$$

$$\boldsymbol{\Psi} = \left(\mu\boldsymbol{I} + 2\widehat{\beta}_1\boldsymbol{L} + 2\widehat{\beta}_2\boldsymbol{L}^2\right)^{-1} \mu\widehat{\boldsymbol{\Omega}}^t = \left(\boldsymbol{I} + 2\beta_1\boldsymbol{L} + 2\beta_2\boldsymbol{L}^2\right)^{-1} \widehat{\boldsymbol{\Omega}}^t,$$

where for simplicity we absorbed $\mu$ inside $\beta_i$, i.e., $\{\beta_1, \beta_2\} = \{\frac{\widehat{\beta}_1}{\mu}, \frac{\widehat{\beta}_2}{\mu}\}$. Defining $H(\boldsymbol{L}; \beta_1, \beta_2) = \left(\boldsymbol{I} + 2\beta_1\boldsymbol{L} + 2\beta_2\boldsymbol{L}^2\right)^{-1}$ to be the designed filter, the proposed graph filter is derived as

$$\widehat{\boldsymbol{\Psi}}^t = \boldsymbol{\Psi} = H(\boldsymbol{L}; \beta_1, \beta_2)\widehat{\boldsymbol{\Omega}}^t. \tag{11}$$

$H(\boldsymbol{L}; \beta_1, \beta_2)$ can also be expressed as $H(\boldsymbol{L}; \beta_1, \beta_2) = \boldsymbol{V} h(\boldsymbol{\Lambda}; \beta_1, \beta_2)\boldsymbol{V}^T$ where $\boldsymbol{V} \in \mathbb{R}^{K \times K}$ and $\boldsymbol{\Lambda} = \text{diag}[\lambda_1, ..., \lambda_K] \in \mathbb{R}^{K \times K}$ are the eigenvector and eigenvalue matrix from the eigenvalue decomposition of $\boldsymbol{A}$ (Stanković et al., 2019). Additionally, the filter operator $h(\boldsymbol{\Lambda}; \beta_1, \beta_2)$ on a specific eigenvalue can be further specified as

$$h(\lambda_k; \beta_1, \beta_2) = \frac{1}{1 + \beta_1\lambda_k + \beta_2\lambda_k^2}, \tag{12}$$

where $\lambda_k$ is called a graph frequency in the graph signal processing (GSP) community (Ortega et al., 2018). In equation 12, $h(\cdot)$ is a GSP-inspired low-pass filter tunable by $\beta_1$ and $\beta_2$. An illustration of such a filter is depicted in Section A.6.3. The second term, executed on the server, relies on access to all device graph signals during optimization. Figure 1 illustrates FEDPNP's mechanism, and Algorithm 1 outlines the proposed FEDPNP algorithm.

**Remark 1:** The second term in $(\mathcal{P}4)$ functions as a versatile graph denoiser that can adapt to various denoising settings by incorporating a noise control parameter. This aligns with the Plug and Play (PnP) prior commonly used in inverse problems. Like PnP algorithms in inverse problems, where denoising strength decreases during optimization (Rasti-Meymandi et al., 2023), our proposed denoiser's parameters can transition from high $\beta_i$ values (strong denoising) to lower values (weaker denoising). To achieve this, we introduce mediator parameters $\nu_1^t$ and $\nu_2^t$ that decay exponentially with a rate of $\nu^{(0)}(1-\eta)^t$, where $\nu^{(0)}$ is the initial denoising parameter and $\eta$ is the denoising decay rate. Consequently, the denoising strength of $H(\boldsymbol{L}; \nu_1^t, \nu_2^t)$ gradually diminishes until it matches $H(\boldsymbol{L}; \beta_1, \beta_2)$, where $\beta_1, \beta_2 \le \nu^{(0)}$.

**Remark 2:** We further augment the graph denoising in equation 11 to generalize our framework. The augmented solution to the second subproblem is

$$\widehat{\boldsymbol{\Psi}}^t = \boldsymbol{\Psi} = H(\boldsymbol{L}; \beta_1, \beta_2)\text{diag}[\zeta_1, ..., \zeta_K]\widehat{\boldsymbol{\Omega}}^t, \tag{13}$$

where $\text{diag}[\zeta_1, ..., \zeta_K]$ is the coefficient matrix, similar to the weights in the FedAvg algorithm. Note that $\zeta_k$, $k = 1, ..., K$, privileges each participant individually without any consideration of their graph connectivity. Thus, when denoiser parameters $\beta_1 = \beta_2 \longrightarrow \infty$, the second subproblem approximates a uniform weighted average of the graph signal across all nodes. As $\beta_1 = \beta_2 \longrightarrow \infty$, $h(\lambda; \infty, \infty) = 1$ when $\lambda = 0$ and $h(\lambda; \infty, \infty) = 0$ for all other $\lambda$. The addition of $\frac{\mu}{2}||\boldsymbol{\Psi}^{t-1}[k] - \boldsymbol{\Omega}||_2^2$, with $\boldsymbol{\Psi}^{t-1}[k]$ representing the weighted average of all model weights for $k = 1, ..., K$ (i.e., $\boldsymbol{\Psi}^{t-1}[k] = \boldsymbol{\omega}_G^{t-1}$), transforms FEDPNP into FedProx through regularization, covering both FedAvg and FedProx. More insights on graph denoising and FEDPNP generalization can be found in A.6.2.

---

**Algorithm 1:** FEDPNP with dynamic denoising parameter.

---

Input: $R$, $\beta_1$, $\beta_2$, $\nu^0$ $\eta$, $D_k$, $\boldsymbol{\omega}_k^0$,
```
-Initialization:  Local Processing Step
```
gather statistical information on $D_k$, $k = 1 : K$ and send them to the server
```
-Initialization:  Server Processing Step
```
create the adjacency matrix $\boldsymbol{A}$ based on participants' similarity using the statistical information stack $\boldsymbol{\omega}_k^0$ to create $\boldsymbol{\Psi}^0$
**for** *communication round* $t = 1 : R$ **do**

    ```/* Local Processing Step (Psuedo-projection step)          */```

    **for** *all* $k = 1 : K$ *in parallel* **do**

        $\boldsymbol{\omega}_k^t \longleftarrow \boldsymbol{\Psi}^{t-1}[k]$

        **for** *each local update* $r = 1 : E$ **do**

            $\mathcal{B}_k \longleftarrow$ create mini-batches from $D_k$

            **for** *each mini-batches* $b_i \in \mathcal{B}_k$ **do**

                $\widehat{\boldsymbol{\omega}}_k^t \longleftarrow$ minimize $f_k(\boldsymbol{\omega}_k^t) + \frac{1}{2}||\boldsymbol{\omega}_k^t - \boldsymbol{\Psi}^{t-1}[k]||$

            **end**

        **end**

    **end**

    ```/* Server Processing Step (Denoising step)                 */```

    collect $\widehat{\boldsymbol{\omega}}_k^t$ and stack to create $\widehat{\boldsymbol{\Omega}}^t$

    $\boldsymbol{\Psi}^{(t)} \longleftarrow H(\boldsymbol{L}; \nu_1^t, \nu_2^t)\text{diag}[\zeta_1, ..., \zeta_K]\widehat{\boldsymbol{\Omega}}^t$ equation 13, where $\zeta_k = \frac{|D_k|}{\sum_{k=1}^K |D_k|}$

    **if** $\nu_1^t \geq \beta_1$ **then** $\nu_1^t \longleftarrow \nu^0(1 - \eta)^t$ **else** $\nu_1^t \longleftarrow \beta_1$

    **if** $\nu_2^t \geq \beta_2$ **then** $\nu_2^t \longleftarrow \nu^0(1 - \eta)^t$ **else** $\nu_2^t \longleftarrow \beta_2$

**end**

---

## 3    CONVERGENCE ANALYSIS

In this section, we analyze the convergence of FEDPNP in scenarios involving both convex and non-convex problems. To establish the theoretical foundation for our assessment of FL algorithms, we adopt fundamental assumptions commonly found in (Li et al., 2020a;b; Setayesh et al., 2022; Li et al., 2019), forming the basis of our analytical framework.

**Assumption 1** *(L-Lipschitz continuous gradient) The local loss function $f_k(\cdot)$ is L-Lipschitz continuous gradient for all clients $k \in \{1.2, \cdots, K\}$, meaning that $\forall \boldsymbol{\omega}$ and $\hat{\boldsymbol{\omega}}$, the inequality $\|\nabla f_k(\boldsymbol{\omega}) - \nabla f_k(\hat{\boldsymbol{\omega}})\| \leq L\|\boldsymbol{\omega} - \hat{\boldsymbol{\omega}}\|$ holds with some $L > 0$.*

**Assumption 2** *(Strong convexity) For each client $k$, there exists $\alpha > 0$ such that $\nabla^2 f_k(\cdot) \succeq -\alpha \mathbf{I}$, with $\mathbf{I}$ as the identity matrix. Consequently, this implies that the function $l_k(\boldsymbol{\omega}, \boldsymbol{\Psi}^t[k])$ in equation 9 is $\alpha'$-strongly convex, where $\alpha' := -\alpha + \mu > 0$, signifying that $\forall \boldsymbol{\omega}$ and $\hat{\boldsymbol{\omega}}$, the inequality $\|\nabla l_k(\boldsymbol{\omega}, \boldsymbol{\Psi}^t[k]) - \nabla l_k(\hat{\boldsymbol{\omega}}, \boldsymbol{\Psi}^t[k])\| \geq \alpha'\|\boldsymbol{\omega} - \hat{\boldsymbol{\omega}}\|$ holds.*

**Assumption 3** *($\gamma_k^t$-inexact solution) Local updates result in a $\gamma_k^t$-inexact solution $\boldsymbol{\omega}_k^{t+1}$ for $min_{\boldsymbol{\omega}} l_k(\boldsymbol{\omega}, \boldsymbol{\Psi}^t[k])$ for every $k$ and $t$, where $\|\nabla l_k(\boldsymbol{\omega}_k^{t+1}, \boldsymbol{\Psi}^t[k])\| \leq \gamma_k^t \|\nabla l_k(\boldsymbol{\Psi}^t[k], \boldsymbol{\Psi}^t[k])\|$. We assume $\gamma_k^t \in [0, 1]$, where $\gamma_k^t = 0$ signifies optimality, with smaller $\gamma_k^t$ indicating higher accuracy.*

Now, we will explore the convergence of FEDPNP in a non-convex setting, as described in the following theorem, considering a single step of FEDPNP.

**Theorem 1** *(Non-convex FEDPNP convergence) Let Assumptions 1-3 hold, and $\boldsymbol{\omega}_k^t$ not be a stationary solution for client $k$ at step $t$. Provided that $\mu$, $\alpha$, $\gamma_k^t$, and $L$ are chosen such that*

$$\upsilon_k^t = \frac{1}{\mu} - \frac{\gamma_k^t}{\mu} - \frac{L(1+\gamma_k^t)}{\mu\alpha'} - \frac{L(1+\gamma_k^t)^2}{2\alpha'^2} > 0, \tag{14}$$

*then, at iteration $t$ within Alg. 1, we observe the following decrease in each local loss function:*

$$f_k(\boldsymbol{\omega}_k^{t+1}) \leq f_k(\boldsymbol{\Psi}^t[k]) - \upsilon_k^t \|\nabla f_k(\boldsymbol{\Psi}^t[k])\|^2. \tag{15}$$

*The proof of Theorem 1 is provided in the supplementary material.*

We can employ the established sufficient decrease condition in Theorem 1 to elucidate the convergence rate of FEDPNP when applied to non-convex functions $f_k(\cdot)$.

**Theorem 2** *(Convergence rate: FEDPNP) Let the assumption of Theorem 1 be met during each iteration of FEDPNP, and $\delta_k := \sum_{t=0}^{T-1} f_k(\boldsymbol{\Psi}^t[k]) - f_k(\boldsymbol{\omega}_k^{t+1})$. It follows that after $T = \mathcal{O}\left(\frac{\delta_k}{\epsilon\upsilon_k}\right)$ iterations of FEDPNP, we have $\frac{1}{T}\sum_{t=0}^{T-1} \|\nabla f_k(\boldsymbol{\Psi}^t[k])\|^2 \leq \epsilon$, with $\upsilon_k = \min_t \upsilon_k^t$. The proof of Theorem 2 is provided in the supplementary material.*

Although the findings presented so far are applicable to non-convex functions $f_k(\cdot)$, we also analyze the convergence behavior for the specific scenario of convex loss functions.

**Corollary 1** *(FEDPNP convergence: Convex case) Let Assumption 1 hold, $f_k(\cdot)$ be convex, and $\gamma_k^t = 0$ for any $k$ and $t$ (indicating precise solutions to all local problems). In such a case, given $\mu > 1.5L$, we can get that $\upsilon = \frac{1}{\mu} - \frac{3L}{2\mu^2}$ and $f_k(\boldsymbol{\omega}_k^{t+1}) \leq f_k(\boldsymbol{\Psi}^t[k]) - \upsilon \|\nabla f_k(\boldsymbol{\Psi}^t[k])\|^2$. Consequently, it can be deduced that after conducting $T = \mathcal{O}\left(\frac{L\delta_k}{\epsilon}\right)$ iterations of the FEDPNP method, we can stablish that $\frac{1}{T}\sum_{t=0}^{T-1} \|\nabla f_k(\boldsymbol{\Psi}^t[k])\|^2 \leq \epsilon$, with $\delta_k := \sum_{t=0}^{T-1} f_k(\boldsymbol{\Psi}^t[k]) - f_k(\boldsymbol{\omega}_k^{t+1})$. You can find the proof of Corollary 1 in the supplementary materials.*

## 4 EXPERIMENTAL RESULTS

In this section, we experiment with FEDPNP across diverse datasets, analyzing parameter influence ($\beta$ and $\mu$) on convergence in classification tasks. We highlight the versatility of denoising techniques for the second subproblem in equation 8, avoiding explicit closed-form derivations.

### 4.1 EXPERIMENTAL SETTINGS

**Datasets and ML Models:** We utilize MNIST and CIFAR-10 datasets for image classification in FL (LeCun et al., 1998; Krizhevsky et al., 2009). Participants receive local datasets with label distribution based on a Dirichlet distribution ($Dir(\kappa)$) (Yurochkin et al., 2019; Wang et al., 2020). We test two $\kappa$ values, 0.2 and 0.5, to highlight varying data heterogeneity. Additionally, we extend our experiments to Human Activity Recognition (HAR), a real-world classification task. HAR datasets are chosen for two reasons: they inherently raise privacy concerns, making them ideal for FL solutions, and they naturally exhibit data heterogeneity, eliminating the need for synthetic scenario simulations like in MNIST and CIFAR-10. Specifically, we use the HARBOX dataset (Ouyang et al., 2022), comprising daily activities from 121 users (aged 17–55), resampled at 50 Hz with 2-second windows, resulting in 900 feature vectors. Importantly, this dataset was collected using 77 different smartphone models, introducing feature skew and data heterogeneity—a perfect setting for our federated experiments. Local datasets are split into 75% training and 25% test sets.

Following the models used in (T Dinh et al., 2020), we employ a 2-layer Conv2D followed by 1 and 2 Fully Connected (FC) layers for MNIST and CIFAR-10 datasets, respectively. In the case of HARBOX, we opt for a fully connected network featuring two FC layers similar to the model used in (Ouyang et al., 2022). All trainable layers undergo batch normalization. Our focus is on contrasting the efficiencies of different frameworks using identical ML models, rather than emphasizing specific architectures for state-of-the-art numerical outcomes. More details are available in Section A.4.

Table 1: Mean accuracy comparison ($\pm$ std) over $K = 20$ local models under data heterogeneity on CIFAR-10, MNIST, and HARBOX datasets after $R = 400$, with $h$ and $s$ superscripts on FEDPNP denoting the hard thresholding and soft thresholding denoisers defined in 16 and 12, respectively.

| Algorithm | CIFAR-10 | | MNIST | | HARBOX |
|---|---|---|---|---|---|
| | $Dir(0.2)$ | $Dir(0.5)$ | $Dir(0.2)$ | $Dir(0.5)$ | |
| Local Training | 60.07±17.5 | 59.03±16.8 | 96.87±2.2 | 94.07±3.8 | 60.57±22.1 |
| FedAvg | 50.23±20.2 | 53.93±17.5 | 97.87±1.5 | 96.87±2.5 | 67.22±20.4 |
| FedProx | 52.85±18.2 | 54.53±15.2 | 97.47±1.5 | 96.65±2.4 | 69.17±18.5 |
| FedBN | 53.43±17.8 | 54.76±15.1 | 97.76±1.5 | 96.85±2.4 | 68.47±18.7 |
| Per-FedAvg | 61.77±14.6 | 61.03±11.8 | 97.55±1.6 | 96.33±2.5 | 81.02±17.5 |
| pFedMe | 64.45±12.6 | 62.56±11.6 | 97.83±1.4 | 96.74±2.2 | 85.60±15.2 |
| FedPnP$_h$($\mu = 0, \tau = K$) | 60.03±17.5 | 59.23±16.7 | 96.58±2.3 | 94.11±3.8 | 60.50±22.1 |
| FedPnP$_s$($\mu = 0, \beta = 0$) | 60.10±17.5 | 59.10±16.7 | 96.83±2.3 | 94.06±3.8 | 60.55±22.0 |
| FedPnP$_h$($\mu = 0.2, \tau = 1$) | 50.86±18.5 | 52.26±15.4 | 97.30±1.7 | 96.50±2.2 | 67.18±18.6 |
| FedPnP$_h$($\mu = 0.2, \tau = *$) | 69.88±11.7(*$\tau = 15$) | 62.61±12.2(*$\tau = 16$) | 97.35±1.8(*$\tau = 2$) | **97.23**±1.4(*$\tau = 2$) | 81.83±17.7(*$\tau = 5$) |
| FedPnP$_s$($\mu = 0.2, \beta = 0.005$) | 71.33±11.1 | 67.02±10.3 | 98.30±1.3 | 96.90±1.6 | 86.15±12.6 |
| FedPnP$_s$($\mu = 0.2, \beta = 0.0005$) | **72.04**±**10.0** | **68.98**±**9.2** | **98.43**±**1.3** | 97.02±1.8 | **90.55**±**10.6** |

**Implemenattion Details:** In all experiments, we maintain a consistent learning rate of $0.01$ with an exponential weight decay rate of $0.96$. Local updates are performed for a fixed number of $E = 5$ iterations. Without loss of generality, we assume $\beta_1 = \beta_2 = \beta$ for simplicity. We configure the number of participants as $K = 20$ for MNIST and CIFAR-10 and $K = 40$ for HARBOX. All model weights initialize uniformly as $\boldsymbol{\omega}_k^0 = \boldsymbol{\omega}_0$ for $k = 1, ..., K$. Unless specified otherwise, we set the denoising decay rate to $\eta = 0.1$, the initial denoising parameter to $\nu^{(0)} = 1$, and $\mu = 0.2$. Note that better models result in higher test accuracy. All experiments are the averages of five independent runs conducted on a single GPU, the GeForce RTX 2080 Ti, with 128 GB of RAM, utilizing the PyTorch library (Paszke et al., 2017). Further implementation details are provided in A.4.3.

We also explore an alternative denoiser for the second subproblem in equation 8 within Algorithm 1 to assess FEDPNP's "*Plug and Play*" nature. The newly defined denoiser is as follows:

$$h(\lambda; \tau) = \begin{cases} 1 & \text{if the index of } \lambda < \tau \\ 0 & \text{otherwise} \end{cases}, \tag{16}$$

where $\tau$ is an integer value specifying how many eigenvalues of the graph signal should be retained. A smaller $\tau$ yields a smoother graph signal. For example, when $\tau = 1$, the filter retains only the first eigenvalue of the graph's adjacency matrix, representing the signal mean. In another view, the filter in equation 16 is a *hard thresholding* while the one in equation 12 is a *soft thresholding* graph filter.

## 4.2 NUMERICAL RESULTS

We compare FEDPNP with state-of-the-art algorithms like FedProx (Li et al., 2020a), FedBN (Li et al., 2020b), PFedMe (T Dinh et al., 2020), and Per-FedAvg (Fallah et al., 2020) to address data heterogeneity and PFL. Extensive hyperparameter optimization through grid search maximizes their test accuracy. Table 1 shows results across various datasets and heterogeneity levels. Remarkably, FEDPNP ($\beta = 0.0005$) consistently outperforms others, benefiting from the graph-structured participant relationships embedded in our algorithm. Setting $\beta = 0$ and $\tau = K$ in FEDPNP leads to fully local training, with the denoiser having no role in information sharing. Furthermore, FEDPNP's test accuracy closely matches that of FedProx and FedAvg when $\tau = 1$, affirming the generalization of the FEDPNP framework. Comparing hard and soft thresholding versions of FEDPNP (i.e., FEDPNP$_h$ and FEDPNP$_s$, resp.) underscores the denoiser's vital role, with the latter achieving superior accuracy. Hence, designing the denoiser is pivotal for FEDPNP's model performance.

## 4.3 EFFECT OF DIFFERENT PARAMETERS/DENOISERS ON CONVERGENCE CURVE

**Effect of $\beta$:** In this section, we examine the impact of the smoothing strength parameter, $\beta$, outlined in equation equation 12. Figure 2 demonstrates FEDPNP's convergence across diverse datasets with varying $\beta$ values. Smaller $\beta$ consistently leads to higher test accuracy, facilitating more effective structured information sharing according to the graph $\mathcal{G}$. FEDPNP with the recommended $\beta$ generally outperforms both FedAvg and FedProx, as seen in improved training losses across communication rounds. The method excels by leveraging the fine-grained interconnection of local models, controlled by the parameter $\beta$. While FedProx shows faster loss reduction compared to FEDPNP with $\beta = 1, 0.5, 0.05$ in Figure 2e, its corresponding test accuracy (Figure 2b) remains lower. Larger $\beta$ values result in smoother graph signal behavior, resembling FedAvg with $\mu = 0$ and FedProx with $\mu > 0$. However, setting $\beta = 0$ causes

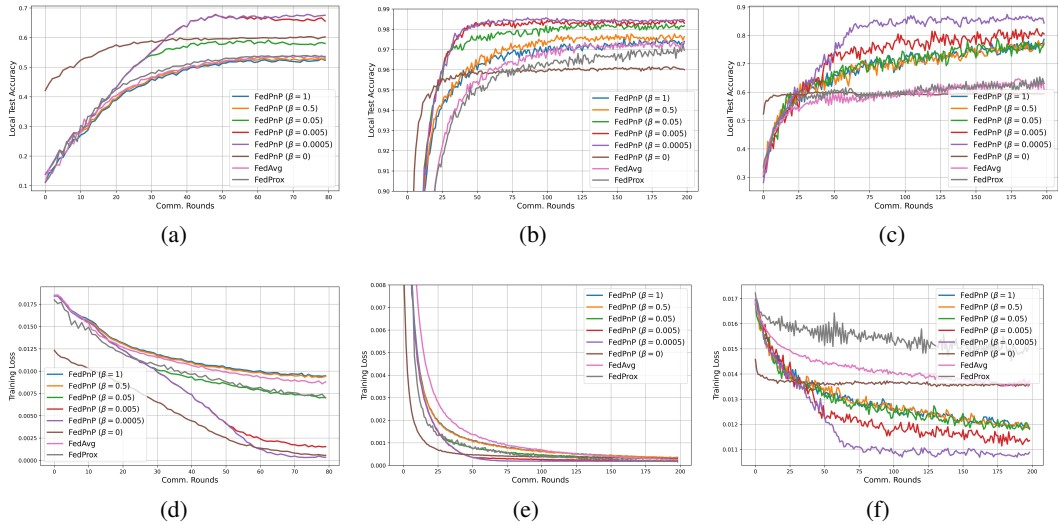

(a)                          (b)                          (c)

(d)                          (e)                          (f)

Figure 2: Effect of $\beta$ on the convergence of FEDPNP compared to FedAvg and FedProx when $K = 20$, $\mu = 0.2$, $E = 5$. $a, d$) CIFAR-10 with $Dir(0.5)$, $b, e$) MNIST with $Dir(0.2)$, and $c, f$) HARBOX datasets. For $\beta = 0$, $\nu^{(0)} = 0$ and for the rest, it is $\nu^{(0)} = 1$.

a drop in test accuracy as FEDPNP converges to full local training without knowledge-sharing due to the denoiser not performing graph smoothing. In this case, $\nu^{(0)} = 0$ is also set to emulate such a scenario. Section A.7 presents additional experiments on the effect of $\beta$, model generalization over an out-of-distribution test set, the impact of $E$, and varying batch sizes.

**Effect of** $\mu$**:** We investigate the impact of $\mu$, the regularization parameter, in the first sub-problem of equation 8. With $\beta = 1$ and other parameters fixed, Figure 3 shows its effect on CIFAR-10 and HARBOX datasets. Increasing $\mu$ slows down convergence, but it enhances test accuracy compared to smaller $\mu$ values, revealing a speed-accuracy trade-off.

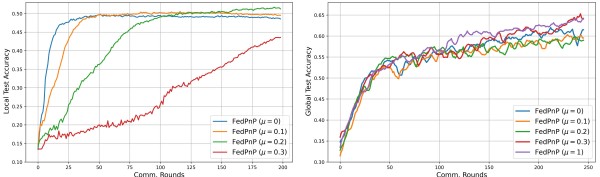

Figure 3: Effect of $\mu$ on the convergence of FEDPNP when $\beta = 1$, $E = 5$. (Right) CIFAR-10 with $Dir(0.5)$ and $K = 20$ and (Left) HARBOX with $K = 40$.

**Effect of denoiser:** Figure 4 shows the new denoiser's performance on the CIFAR-10 and MNIST datasets while varying $\tau$. In Algorithm 1, we incrementally increase $\nu_{(t)}$ over the communication rounds until it reaches $\tau$. Increasing $\tau$ improves test accuracy, similar to reducing $\beta$ in equation 12. This alternate denoiser effectively replaces the second sub-problem in equation 8 without explicit solving, allowing for plugging and playing with various denoisers, akin to PnP algorithms in inverse problems.

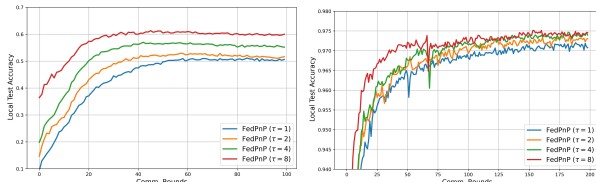

Figure 4: The effect of the plugged-in denoiser defined in equation 16 over defferent $\tau$ on (Right) CIFAR-10 and (Left) MNIST with $Dir(0.5)$ when $\mu = 0.2$, $E = 5$, and $K = 20$.

## 5 CONCLUSION

We introduced FEDPNP, a graph-based Federated Learning (FL) algorithm inspired by inverse problem solutions. Connecting our optimization problem to established inverse problems, we utilized HQS to split it into two parts: data fidelity (local update) with a novel proximal term and a sparsity-inducing term (graph denoiser) adaptable to different denoisers with controlled noise. Our experiments showed FEDPNP's superior performance over other FL algorithms. Future works can explore alternative inverse problem approaches, denoiser roles, graph construction criteria, and privacy concerns. Our current work assumes similar local model architectures, left for future investigation.

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

## A SUPPLEMENTARY MATERIALS

### A.1 PROOF OF THEOREM 1

From the L-Lipschitz continuous gradient property of local loss function $f_k(\cdot)$ (Assumption 1), we have

$$f_k(\boldsymbol{\omega}_k^{t+1}) \leq f_k(\boldsymbol{\Psi}^t[k]) + \langle \nabla f_k(\boldsymbol{\Psi}^t[k]), (\boldsymbol{\omega}_k^{t+1} - \boldsymbol{\Psi}^t[k]) \rangle + \frac{L}{2} \left\| \boldsymbol{\omega}_k^{t+1} - \boldsymbol{\Psi}^t[k] \right\|^2, \quad (17)$$

where $\langle \cdot, \cdot \rangle$ indicates the inner product operation. Then, we shall proceed to independently establish upper bound limits for the last two expressions on the right-hand side of equation 17 in a subsequent manner:

- **Upper bound for** $\left\| \boldsymbol{\omega}_k^{t+1} - \boldsymbol{\Psi}^t[k] \right\|$: Let assume $\tilde{\boldsymbol{\omega}}_k^{t+1} = \arg\min_{\boldsymbol{\omega}} l_k(\boldsymbol{\omega}; \boldsymbol{\Psi}^t[k])$ such that $\nabla l_k(\tilde{\boldsymbol{\omega}}_k^{t+1}; \boldsymbol{\Psi}^t[k]) = 0$. Given the $\alpha'$-convexity property of $l_k(\boldsymbol{\omega}; \boldsymbol{\Psi}^t[k])$ (Assumption 2), and considering the $\gamma_k^t$-inexactness exhibited by each local server for $\boldsymbol{\omega}_k^{t+1}$ (Assumption 3), we can derive the following:

$$\left\| \boldsymbol{\omega}_k^{t+1} - \tilde{\boldsymbol{\omega}}_k^{t+1} \right\| \leq \frac{1}{\alpha'} \left\| \nabla l_k(\boldsymbol{\omega}_k^{t+1}; \boldsymbol{\Psi}^t[k]) - \nabla l_k(\tilde{\boldsymbol{\omega}}_k^{t+1}; \boldsymbol{\Psi}^t[k]) \right\| = \frac{1}{\alpha'} \left\| \nabla l_k(\boldsymbol{\omega}_k^{t+1}; \boldsymbol{\Psi}^t[k]) \right\|$$

$$\leq \frac{1}{\alpha'} \gamma_k^t \left\| \nabla l_k(\boldsymbol{\Psi}^t[k]; \boldsymbol{\Psi}^t[k]) \right\| = \frac{\gamma_k^t}{\alpha'} \left\| \nabla f_k(\boldsymbol{\Psi}^t[k]) \right\|. \quad (18)$$

Likewise, we can deduce the following due to $\alpha'$-convexity of $l_k(\boldsymbol{\omega}; \boldsymbol{\Psi}^t[k])$ (Assumption 2):

$$\left\| \tilde{\boldsymbol{\omega}}_k^{t+1} - \boldsymbol{\Psi}^t[k] = \right\| \leq \frac{1}{\alpha'} \left\| \nabla l_k(\tilde{\boldsymbol{\omega}}_k^{t+1}; \boldsymbol{\Psi}^t[k]) - \nabla l_k(\boldsymbol{\Psi}^t[k]; \boldsymbol{\Psi}^t[k]) \right\|$$

$$= \frac{1}{\alpha'} \left\| \nabla l_k(\boldsymbol{\Psi}^t[k]; \boldsymbol{\Psi}^t[k]) \right\| = \frac{1}{\alpha'} \left\| \nabla f_k(\boldsymbol{\Psi}^t[k]) \right\|. \quad (19)$$

Therefore, when we put together equation 18 and equation 19 using the triangle inequality, we obtain:

$$\left\| \boldsymbol{\omega}_k^{t+1} - \boldsymbol{\Psi}^t[k] \right\| \leq \left\| \boldsymbol{\omega}_k^{t+1} - \tilde{\boldsymbol{\omega}}_k^{t+1} \right\| + \left\| \tilde{\boldsymbol{\omega}}_k^{t+1} - \boldsymbol{\Psi}^t[k] \right\|$$

$$\leq \frac{\gamma_k^t}{\alpha'} \left\| \nabla f_k(\boldsymbol{\Psi}^t[k]) \right\| + \frac{1}{\alpha'} \left\| \nabla f_k(\boldsymbol{\Psi}^t[k]) \right\| = \frac{(\gamma_k^t + 1)}{\alpha'} \left\| \nabla f_k(\boldsymbol{\Psi}^t[k]) \right\|. \quad (20)$$

- **Upper bound for** $\langle \nabla f_k(\boldsymbol{\Psi}^t[k]), (\boldsymbol{\omega}_k^{t+1} - \boldsymbol{\Psi}^t[k]) \rangle$: We can rewrite $(\boldsymbol{\omega}_k^{t+1} - \boldsymbol{\Psi}^t[k])$ as:

$$\boldsymbol{\omega}_k^{t+1} - \boldsymbol{\Psi}^t[k] = -\frac{1}{\mu} \nabla f_k(\boldsymbol{\Psi}^t[k]) + \frac{1}{\mu}(\nabla f_k(\boldsymbol{\Psi}^t[k]) - \nabla f_k(\boldsymbol{\omega}_k^{t+1})) + \frac{1}{\mu}(\nabla f_k(\boldsymbol{\omega}_k^{t+1}) + \mu(\boldsymbol{\omega}_k^{t+1} - \boldsymbol{\Psi}^t[k]))$$

$$= -\frac{1}{\mu} \nabla f_k(\boldsymbol{\Psi}^t[k]) + \frac{1}{\mu}(\nabla f_k(\boldsymbol{\Psi}^t[k]) - \nabla f_k(\boldsymbol{\omega}_k^{t+1})) + \frac{1}{\mu} \nabla l_k(\boldsymbol{\omega}_k^{t+1}; \boldsymbol{\Psi}^t[k]).$$

Hence, the term $\langle \nabla f_k(\boldsymbol{\Psi}^t[k]), (\boldsymbol{\omega}_k^{t+1} - \boldsymbol{\Psi}^t[k]) \rangle$ will be written as

$$\left\langle \nabla f_k(\boldsymbol{\Psi}^t[k]), -\frac{1}{\mu} \nabla f_k(\boldsymbol{\Psi}^t[k]) + \frac{1}{\mu}(\nabla f_k(\boldsymbol{\Psi}^t[k]) - \nabla f_k(\boldsymbol{\omega}_k^{t+1})) + \frac{1}{\mu} \nabla l_k(\boldsymbol{\omega}_k^{t+1}; \boldsymbol{\Psi}^t[k]) \right\rangle =$$

$$-\frac{1}{\mu} \left\| \nabla f_k(\boldsymbol{\Psi}^t[k]) \right\|^2 + \frac{1}{\mu} \left\langle \nabla f_k(\boldsymbol{\Psi}^t[k]), \nabla f_k(\boldsymbol{\Psi}^t[k]) - \nabla f_k(\boldsymbol{\omega}_k^{t+1}) \right\rangle + \frac{1}{\mu} \left\langle \nabla f_k(\boldsymbol{\Psi}^t[k]), \nabla l_k(\boldsymbol{\omega}_k^{t+1}; \boldsymbol{\Psi}^t[k]) \right\rangle,$$

where, by applying the Cauchy-Schwarz inequality on the last two terms, we can establish the following upper bound as

$$\left\langle \nabla f_k(\boldsymbol{\Psi}^t[k]), (\boldsymbol{\omega}_k^{t+1} - \boldsymbol{\Psi}^t[k]) \right\rangle \leq -\frac{1}{\mu} \left\| \nabla f_k(\boldsymbol{\Psi}^t[k]) \right\|^2 + \frac{1}{\mu} \left\| \nabla f_k(\boldsymbol{\Psi}^t[k]) \right\| \left\| \nabla f_k(\boldsymbol{\Psi}^t[k]) - \nabla f_k(\boldsymbol{\omega}_k^{t+1}) \right\|$$

$$+ \frac{1}{\mu} \left\| \nabla f_k(\boldsymbol{\Psi}^t[k]) \right\| \left\| \nabla l_k(\boldsymbol{\omega}_k^{t+1}; \boldsymbol{\Psi}^t[k]) \right\|.$$

Based on the L-Lipschitz continuity of $f_k(\cdot)$ and $\gamma_k^t$-exactness of $l_k(\boldsymbol{\omega}, \boldsymbol{\Psi}^t[k])$ for $\boldsymbol{\omega}_k^{t+1}$, we have

$$\left\langle \nabla f_k(\boldsymbol{\Psi}^t[k]), (\boldsymbol{\omega}_k^{t+1} - \boldsymbol{\Psi}^t[k]) \right\rangle \leq -\frac{1}{\mu} \left\| \nabla f_k(\boldsymbol{\Psi}^t[k]) \right\|^2 + \frac{L}{\mu} \left\| \nabla f_k(\boldsymbol{\Psi}^t[k]) \right\| \left\| \boldsymbol{\omega}_k^{t+1} - \boldsymbol{\Psi}^t[k] \right\|$$

$$+ \frac{\gamma_k^t}{\mu} \left\| \nabla f_k(\boldsymbol{\Psi}^t[k]) \right\| \left\| \nabla f_k(\boldsymbol{\Psi}^t[k]) \right\|,$$

where the second term on the right-hand side can be upper bound by equation 20, resulting in the following

$$\left\langle \nabla f_k(\mathbf{\Psi}^t[k]), (\boldsymbol{\omega}_k^{t+1} - \mathbf{\Psi}^t[k]) \right\rangle \leq \left( \frac{L(1+\gamma_k^t)}{\mu\alpha'} + \frac{(\gamma_k^t - 1)}{\mu} \right) \left\| \nabla f_k(\mathbf{\Psi}^t[k]) \right\|^2. \quad (21)$$

Finally, by combining equations 17, 20, and 21, we obtain

$$f_k(\boldsymbol{\omega}_k^{t+1}) \leq f_k(\mathbf{\Psi}^t[k]) + \left( \frac{L(1+\gamma_k^t)}{\mu\alpha'} + \frac{(\gamma_k^t - 1)}{\mu} \right) \left\| \nabla f_k(\mathbf{\Psi}^t[k]) \right\|^2 + \frac{1 + L(\gamma_k^t)^2}{2\alpha'^2} \left\| \nabla f_k(\mathbf{\Psi}^t[k]) \right\|^2, \quad (22)$$

in which we can rewrite it as

$$f_k(\boldsymbol{\omega}_k^{t+1}) \leq f_k(\mathbf{\Psi}^t[k]) - \upsilon_k^t \left\| \nabla f_k(\mathbf{\Psi}^t[k]) \right\|^2, \quad (23)$$

where $\upsilon_k^t = \frac{1}{\mu} - \frac{\gamma_k^t}{\mu} - \frac{L(1+\gamma_k^t)}{\mu\alpha'} - \frac{L(1+\gamma_k^t)^2}{2\alpha'^2}$. Based on the above-mentioned inequality, it can be inferred that by choosing a sufficiently large penalty parameter $\mu$, we can achieve a reduction in the objective value of $f_k(\boldsymbol{\omega}_k^{t+1}) - f_k(\mathbf{\Psi}^t[k])$, and this reduction is directly proportional to $\left\| \nabla f_k(\mathbf{\Psi}^t[k]) \right\|^2$. This completes the proof of Theorem 1.

## A.2 PROOF OF THEOREM 2

Suppose we meet the assumption of Theorem 1 for the client $k$ during each iteration of FedPnP. Then we have the following inequality at every iteration:

$$\left\| \nabla f_k(\mathbf{\Psi}^t[k]) \right\|^2 \leq \frac{f_k(\mathbf{\Psi}^t[k]) - f_k(\boldsymbol{\omega}_k^{t+1})}{\upsilon_k^t}, \quad t \in \{0, 1, \cdots, T-1\} \quad (24)$$

Upon aggregating the above inequality across all $T$ iterations, we will get:

$$\frac{1}{T} \sum_{t=0}^{T-1} \left\| \nabla f_k(\mathbf{\Psi}^t[k]) \right\|^2 \leq \frac{1}{T} \sum_{t=0}^{T-1} \frac{f_k(\mathbf{\Psi}^t[k]) - f_k(\boldsymbol{\omega}_k^{t+1})}{\upsilon_k^t} \overset{(a)}{\leq} \frac{\sum_{t=0}^{T-1} f_k(\mathbf{\Psi}^t[k]) - f_k(\boldsymbol{\omega}_k^{t+1})}{T\upsilon_k} = \frac{\delta_k}{T\upsilon_k}, \quad (25)$$

where $(a)$ follows from $\upsilon_k = \min_t \upsilon_k^t$ and $\delta_k := \sum_{t=0}^{T-1} f_k(\mathbf{\Psi}^t[k]) - f_k(\boldsymbol{\omega}_k^{t+1})$. Hence, in order to produce a solution that satisfies the condition of a squared gradient norm less than $\epsilon$ (i.e., $\frac{1}{T} \sum_{t=0}^{T-1} \|\nabla f_k(\mathbf{\Psi}^t[k])\|^2 \leq \epsilon$), it is necessary to ensure that the number of iterations $T$ is $\mathcal{O}\left(\frac{\delta_k}{\epsilon \upsilon_k}\right)$. This completes the proof of Theorem 2.

## A.3 PROOF OF COROLLARY 1

In the convex scenario, with $\alpha = 0$ and $\alpha' = \mu$, assuming that $\gamma_k^t = 0$ for all $k$ and $t$ (meaning that all local subproblems are accurately solved), we can attain a reduction proportional to $\left\| \nabla f_k(\mathbf{\Psi}^t[k]) \right\|^2$, provided that $\upsilon_k^t = \upsilon = \frac{1}{\mu} - \frac{3L}{2\mu^2} > 0$. Hence, if $\mu > 1.5L$, we can express this as:

$$f_k(\boldsymbol{\omega}_k^{t+1}) \leq f_k(\mathbf{\Psi}^t[k]) - \left( \frac{1}{\mu} - \frac{3L}{2\mu^2} \right) \left\| \nabla f_k(\mathbf{\Psi}^t[k]) \right\|^2. \quad (26)$$

Under the constraint $\mu > 1.5L$ and in conjunction with the proof presented for Theorem 2, it is straightforward to determine that the number of iterations required to obtain at least one solution with a squared gradient norm less than $\epsilon$ is $\mathcal{O}\left(\frac{L\delta_k}{\epsilon}\right)$. This completes the proof of Corollary 1.

## A.4 ML MODELS AND TRAINING SPECIFICS

For reproducibility purposes, the details of the ML models are reported here. Tables 2, 3, 4, show the model architectures used for the MNIST, CIFAR-10, and HARBOX datasets, respectively.

### A.4.1 SPECIFICS ON MNIST AND CIFAR-10 DATASETS

**CIFAR-10 (Krizhevsky et al., 2009)**The CIFAR-10 dataset is a widely utilized collection in the realm of computer vision, primarily designed for tasks related to image classification. It encompasses a total of 60,000 color images, distributed across ten distinct categories, with each category containing precisely 6,000 images. This dataset is further segmented into two subsets: a training set comprising 50,000 images and a testing set containing 10,000 images. Every image within the CIFAR-10 dataset is of dimensions 32x32 pixels and possesses three color channels corresponding to red, green, and blue. The ten classes it encompasses encompass a variety of common objects and animals, such as airplanes, automobiles, birds, cats, deer, dogs, frogs, horses, ships, and trucks.

**MNIST (LeCun et al., 1998)** The MNIST dataset holds a prominent and extensively used position in the realm of machine learning and computer vision. It consists of 28x28 pixel grayscale images depicting handwritten digits (ranging from 0 to 9). MNIST encompasses a grand total of 70,000 images, divided into two primary sets: a training set comprising 60,000 images and a test set containing 10,000 images. Each image within MNIST is a grayscale representation with a single channel, where pixel values span from 0 (representing black) to 255 (representing white), thereby encoding different shades of gray.

To create a heterogeneous dataset using MNIST and CIFAR-10, we utilized and adapted publicly available code as employed in (Li et al., 2022). The original repository can be accessed at [1]. In the case of the CIFAR-10 dataset we constructed, the sizes of local datasets vary within the range of $(810, 3642)$, with corresponding label distributions based on the Dirichlet distribution. For each participant, we employed a batch size of $\mathcal{B}_k = 128$. The same configuration was used for the MNIST dataset, with the exception that we opted for a fixed data size of $|D_k| = 600$ for $k = 1, ..., 20$. We made this choice to investigate the impact of the denoising step in the second subproblem of equation 13. By maintaining a balanced data size, we ensured similar coefficients in $\mathrm{diag}[\zeta_1, ..., \zeta_{20}]$. Consequently, only the graph filter $H(\boldsymbol{L})$ became the focal point.

### A.4.2 SPECIFICS ON HARBOX DATASET

The HARBOX dataset is a comprehensive Human Activity Recognition dataset gathered from smartphone sensors. It comprises five common daily life activities, such as calling, hopping, talking, typing, and walking, performed by approximately 120 participants. HARBOX exhibits inherent data heterogeneity due to two primary reasons:

- Each participant exhibits a unique way of performing these activities.
- Different smartphones capture activity samples differently, influenced by their device specifications.

For our experiments, we randomly selected a subset of $K = 40$ participants from the dataset. Similar to the approach taken with the MNIST dataset, the local dataset sizes were determined based on the original recorded sensor signals. No manipulation was performed to alter the number of data samples in each local dataset. Consequently, we observed a variation in local dataset sizes within the range of $(170, 400)$ samples. Furthermore, the batch size for our experiments was consistently set at 128.

Table 2: Layer details for CNNmnist model used for the benchmark on Mnist dataset.

| Layer | Details |
|-------|---------|
| 1 | Conv2D(1, 16, 8, 2, 2) BN(16), Tanh, MaxPool2D(2, 1) |
| 2 | Conv2D(16, 32, 4, 2, 0) BN(32), Tanh, MaxPool2D(2, 1) |
| 3 | FC(512, 32) BN(32), Tanh |
| 4 | FC(32, 10) |

---

[1]https://github.com/Xtra-Computing/NIID-Bench

Table 3: Layer details for Cifarnet model used for the benchmark on CIFAR-10 dataset.

| Layer | Details |
|---|---|
| 1 | Conv2D(3, 6, 5) |
| | BN(6), ReLU, MaxPool2D(2, 2) |
| 2 | Conv2D(6, 16, 5) |
| | BN(16), ReLU, MaxPool2D(2, 2) |
| 3 | FC(400, 120) |
| | BN(120), ReLU |
| 4 | FC(120, 84) |
| | BN(84), ReLU |
| 5 | FC(84, 10) |

Table 4: Layer details for ML model used for the benchmark on HARBOX dataset.

| Layer | Details |
|---|---|
| 1 | FC(900, 400) |
| | BN(400), ReLU |
| 2 | FC(400, 5) |
| | Softmax |

### A.4.3 MODEL IMPLEMENTATION DETAILS

**FedProx (Li et al., 2020a) & FedAvg:** We utilized a modified implementation code publicly accessible on the official FedProx GitHub repository[2]. Notably, in the work by the author (Li et al., 2020a), a specific value for $\mu$ was not provided; rather, it was indicated that $\mu > 0$ leads to improved results. Consequently, we conducted experiments with different values of $\mu$, specifically $\mu = 0.001, 0.01, 0.1$, in order to identify the optimal $\mu$ parameter. Ultimately, we determined that $\mu = 0.01$ yielded the best results. All other hyperparameters were configured in accordance with the details discussed in Section 4.

**Per-FedAvg (Fallah et al., 2020), pFedMe (T Dinh et al., 2020):** Both methods were implemented using their respective code available on GitHub[3]. In the case of Per-FedAvg, we configured it with $K = 5$, $\beta = 0.01$, and a personal learning rate of $0.1$. For pFedMe, we selected the following settings: $K = 5$, $\lambda = 15$, $\beta = 1.0$, and a personal learning rate of $0.1$, as recommended in their GitHub code. It's important to note that these parameter values and notations align with those used in their research paper. The remaining parameters were set in a manner consistent with the discussion in Section 4.

**FedBN (Li et al., 2020b):** The implementation of FedBN is straightforward. We followed the algorithm provided in the appendix section of the original paper. Specifically, on the server side, we aggregate all model weights except their batch-normalization layers.

### A.5 GRAPH CONSTRUCTION

Participants possess a large number of underlying similarities that can be represented via an adjacency matrix. This connectivity arises from low-level similarities, including dataset similarities or participants' proximity, to high-level interrelationships, such as social interconnection among the participants. In order to capture the interconnections and facilitate the exchange of information among participants, a graph must be meticulously crafted. We represent the graph as a bidirectional structure denoted as $\mathcal{G}(\mathcal{V}, \mathcal{E})$. Here, $\mathcal{V}$ encompasses all participants, serving as the vertices within the graph, while $\mathcal{E}$ corresponds to the ensemble of edge connections that interlink these participants. Importantly, the weight attributed to each edge is explicitly conveyed through the adjacency matrix $\boldsymbol{A}$. Within this matrix, the element $(\boldsymbol{A})_{ij}$ indicates the weight of the connection between participants $i$ and $j$. Given that the entirety of the graph's configuration is delineated by the adjacency matrix, our focus in this section lies primarily on the construction of $\boldsymbol{A}$.

---

[2]https://github.com/litian96/FedProx
[3]https://github.com/CharlieDinh/pFedMe

The process of constructing $\boldsymbol{A}$ offers several avenues, classified into two overarching approaches: the learning-based and the fixed-based paradigms.

- In the learning-based paradigm, the graph's structure evolves concomitantly with the Federated Learning (FL) process itself. As one example, the value of $(\boldsymbol{A})_{ij}$ can be adaptively adjusted in proportion to the similarity in weight distributions between participants $i$ and $j$, either in each batch of FL rounds or in every round, i.e., $(\boldsymbol{A})_{ij} \propto ||\boldsymbol{\omega}_i - \boldsymbol{\omega}_j||_2^2$. A parallel concept has been presented in (Chen et al., 2022), showcasing the potential of this approach.

- Contrastingly, the fixed-based approach involves the predetermination and establishment of the graph using prior information provided by the participants. These data-driven cues can be grounded in a spectrum of factors, such as the proximity of participants in terms of distance (Rasti-Meymandi et al., 2022), the physical attributes of their communication and computation resources, or the resemblance between their respective datasets.

Both approaches can be integrated into the proposed FEDPNP algorithm. However, in this work, we construct $\boldsymbol{A}$ based on dataset similarity. We leave the learning-based approach in our future work.

### A.5.1 DATASET SIMILARITY

Measuring the similarity between datasets has become an emergent part of the ML community (Alvarez-Melis & Fusi, 2020). Such information fosters transferring knowledge between models and gives rise to more formidable transfer learning techniques, data distillation approaches, and model distillation algorithms (Alvarez-Melis & Fusi, 2020). If the datasets are coming from one domain, e.g., images of animals or numbers, Pearson Correlation Coefficient (PCC) or Kullback-Leibler (KL) divergence can be readily exploited to capture the similarity between datasets using the data samples. However, in the case of FL, the datasets are distributed, and the samples are not available in one setting. Therefore, we propose a distributed dataset similarity metric specified as follows:

Let $(\boldsymbol{x}_n^i, y_n^i) \in D_i$, $i = 1, ..., K$, $n = 1, ..., |D_i|$ be the $n^{\text{th}}$ data sample belonging to the $i^{\text{th}}$ participant, where $\boldsymbol{x}_n^i \in \mathbb{R}^M$ is the feature vector and $y_n^i \in \mathbb{N}$ is the class label. We assume $D_i$ is drawn from a distribution $P_i$. More specifically, let $\boldsymbol{x}_n^i$ be a sample of a random vector $\mathbf{x}^i$ that constitutes $P_i$.

Based on such a definition, a set of statistical features is obtained. Here, we choose to calculate an estimate of the first to the fourth-order of the statistical information, i.e., the mean, variance, skewness, and kurtosis of $\mathbf{x}^i$ given by

$$s_1^i = \frac{1}{|D_i|} \sum_{n=1}^{|D_i|} \boldsymbol{x}_n^i, \;\; s_2^i = \frac{1}{|D_i|} \sum_{n=1}^{|D_i|} (\boldsymbol{x}_n^i - s_1^i)^2, \;\; s_3^i = \frac{1}{|D_i|} \sum_{n=1}^{|D_i|} \frac{(\boldsymbol{x}_n^i - s_1^i)^3}{\sqrt{(s_2^i)^3}}, \qquad (27)$$

$$s_4^i = \frac{1}{|D_i|} \sum_{n=1}^{|D_i|} \frac{(\boldsymbol{x}_n^i - s_1^i)^4}{(s_2^i)2},$$

where $s_k^i$ is the $k^{\text{th}}$ statistical feature of participant $i$. Note that, due to the assumption that each feature in $D_i$ is a random variable, more statistical information, such as higher orders, can be obtained. Also, note that this computation occurs on the local side. The statistical features are then transmitted to the server at the initialization step of the proposed FEDPNP algorithm to construct $\boldsymbol{A}$. As a result, participants will only send aggregated statistical information about their datasets, reducing communication overhead and preserving privacy. Additionally, the amount of statistical information can be variable, which sets up a trade-off between data privacy and the accuracy of the similarity metric. Finally, on the server side, the statistical features are compared, and the weights in $\boldsymbol{A}$ as

$$(\boldsymbol{A})_{ij} = \frac{1}{4} \sum_{k=1}^{4} ||s_k^i - s_k^j||. \qquad (28)$$

**Remark 3:** It is worthwhile to note that the transmission of such information occurs once (as opposed to every communication round) and only in the initialization of the training procedure, which

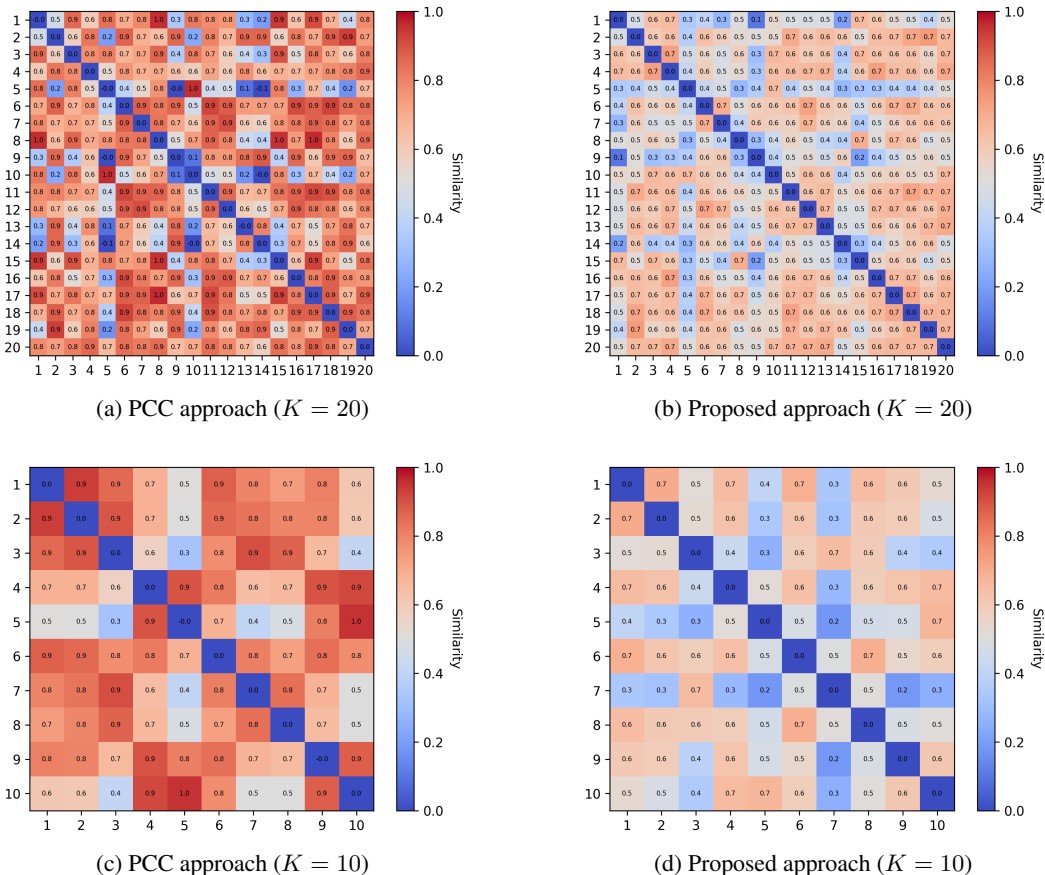

Figure 5: The adjacency matrices constructed by the proposed dataset similarity (b,d) and the PCC approach (a,c) on CIFAR-10 with $Dir(0.5)$ dataset.

will further reduce the likelihood of recurrent data exchanges, thereby ameliorating the potential for prolonged exposure of sensitive information over the course of subsequent communication rounds.

For illustrative purposes, we conduct an experimental demonstration by constructing matrix $\boldsymbol{A}$ using both PCC and the proposed approach on the CIFAR-10 dataset with $Dir(0.5)$. In the context of PCC, we operate under the assumption that all local dataset samples are readily available on the server. These local datasets are generated randomly based on the settings described in Section 4. As shown in Figure 5, the matrices $\boldsymbol{A}$ resulting from the two methods are compared for both $K = 10$ and $K = 20$. Evidently, the proposed distributed similarity metric exhibits a similar correlation with the PCC approach. One advantage of our approach is that it incorporates higher-order statistical attributes within the similarity measurement, thereby enhancing its capability to capture more nuanced dataset similarities. Moreover, the proposed similarity metric also obviates the necessity for complete dataset presence, which is in line with FL algorithms.

## A.6 GRAPH FILTERING AS A DENOISER

To gain insight into the functioning of graph denoisers defined in equations 12 and 16, we provide a brief overview of Graph Signal Processing (GSP). Readers may refer to (Ortega et al., 2018) for a comprehensive detail on GSP. Since both denoisers operate in the Fourier domain, we begin by introducing the concepts of Graph Fourier Transform (GFT) and Inverse GFT (IGFT). Subsequently, we explain how graph filtering is executed using GFT. Lastly, we illustrate how incorporating these denoisers into FEDPNP extends to encompass popular algorithms like FedAvg (McMahan et al., 2017) and FedProx (Li et al., 2020a) through a motivating example.

### A.6.1 GRAPH SIGNAL PROCESSING BACKGROUND

Once the adjacency matrix $\boldsymbol{A}$ is created (e.g., from Section A.5) the whole structure and properties of the graph $\mathcal{G}$ can be defined. Recall that we construct a bidirectional graph. A significant concept in bidirectional graphs is the *combinatorial graph Laplacian*, which is defined as $\boldsymbol{L} = \boldsymbol{D} - \boldsymbol{A}$. In this equation, $\boldsymbol{D}$ represents the diagonal degree matrix of the graph $\mathcal{G}$, with $(\boldsymbol{D})_{ii}$ calculated as the sum of entries in the $i$-th row of $\boldsymbol{A}$. It's important to note that for bidirectional graphs like $\mathcal{G}$, $\boldsymbol{L}$ is always positive semidefinite, and all its eigenvalues are non-negative real values, as demonstrated in (Ortega et al., 2018). As discussed in (Stanković et al., 2019; Ortega et al., 2018), the natural choice is to employ the eigenvectors of the graph Laplacian $\boldsymbol{L}$ as eigenfunctions, forming the basis for the Fourier transform. This leads to the extraction of Fourier bases through the eigenvalue decomposition of the graph Laplacian, expressed as:

$$\boldsymbol{L} = \boldsymbol{V}\boldsymbol{\Lambda}\boldsymbol{V}^T. \tag{29}$$

In this equation, $\boldsymbol{V}$ is a matrix containing the $K$ eigenvectors of $\boldsymbol{L}$, and $\boldsymbol{\Lambda}$ is a diagonal matrix with eigenvalues $\lambda_0$ through $\lambda_{K-1}$. Consequently, if we order the eigenvalues based on the Total Variation (TV) of their corresponding eigenvectors, the eigenvector associated with $\lambda_0$ represents the DC basis, while higher $\lambda_i$ values (for $i = 1, ..., K-1$) correspond to higher frequency bases in the graph.

With this framework in place, we can calculate the GFT and the IGFT of the weight matrix, represented as $GF(\boldsymbol{\Omega}^{(t)})$ and $IGF(GF(\boldsymbol{\Omega}^{(t)}))$:

$$\boldsymbol{\Omega}_{GF}^{(t)} = GF(\boldsymbol{\Omega}^{(t)}) = \boldsymbol{V}^T\boldsymbol{\Omega}^{(t)}, \tag{30}$$

$$\boldsymbol{\Omega}^{(t)} = IGF(\boldsymbol{\Omega}_{GF}^{(t)}) = \boldsymbol{V}\boldsymbol{\Omega}_{GF}^{(t)}. \tag{31}$$

These coefficients, $\boldsymbol{\Omega}_{GF}^{(t)}$, represent the frequency components of the weight matrix. To perform filtering on specific graph frequencies, we follow three steps: (i) GFT, (ii) multiplication of the coefficients by the filter frequency response, and (iii) IGFT of the result. To facilitate this, a graph filter is defined in matrix form, denoted as $H(\boldsymbol{L})$, which can be expressed as $H(\boldsymbol{L}) = \boldsymbol{V}h(\boldsymbol{\Lambda})\boldsymbol{V}^T$, where $h(\boldsymbol{\Lambda})$ is the filter operator defined in equations 12 and 16, with $h(\boldsymbol{\Lambda}) = \mathrm{diag}[h(\lambda_0), ..., h(\lambda_{k-1})]$.

In a more concise form, the filtered weight matrix, which was presented in equation 11 is computed as follows:

$$\widehat{\boldsymbol{\Psi}}^{(t)} = \boldsymbol{\Psi} = H(\boldsymbol{L}; \cdot)\widehat{\boldsymbol{\Omega}}^{(t)} = \boldsymbol{V}h(\boldsymbol{\Lambda})\underbrace{\boldsymbol{V}^T\widehat{\boldsymbol{\Omega}}^{(t)}}_{\text{GFT}} \tag{32}$$

$$= \boldsymbol{V}\underbrace{h(\boldsymbol{\Lambda})\widehat{\boldsymbol{\Omega}}_{GF}^{(t)}}_{\text{frequency response}} \tag{33}$$

$$= \underbrace{\boldsymbol{V}\widehat{\boldsymbol{\Omega}}_{GF}^{(t)}}_{\text{IGFT}}. \tag{34}$$

### A.6.2 GENERALIZATION OF FEDPNP: A MOTIVATING EXAMPLE

To gain a deeper insight into the proposed denoising mechanism called FEDPNP in the frequency domain, we present a simplified example as follows:

Let's consider an undirected light graph with $K = 3$ and an adjacency matrix $\boldsymbol{A}_{3\times 3}$. Each node or participant $i$ possesses a model with a two-variable vector $\boldsymbol{\omega}_i = [\omega_{i1}, \omega_{i2}]$. The Laplacian matrix of $\boldsymbol{A}$ is calculated as

$$\boldsymbol{L} = \boldsymbol{D} - \boldsymbol{A} = \begin{bmatrix} d_1 - a_{11} & -a_{12} & -a_{13} \\ -a_{21} & d_2 - a_{22} & -a_{23} \\ -a_{31} & -a_{32} & d_3 - a_{33} \end{bmatrix}, \tag{35}$$

where $d_i = \sum_{j=1}^{3} a_{ij}$ is the degree of each node. We then decompose $\boldsymbol{L}_{3 \times 3}$ as $\boldsymbol{L} = \boldsymbol{V} \boldsymbol{\Lambda} \boldsymbol{V}^T$ and assume ordered eigenvalues as $\lambda_0 < \lambda_1 < \lambda_2$. Consequently, we can express the eigenvalues and eigenvectors as

$$\boldsymbol{V} = \begin{bmatrix} 1 & v_{12} & v_{13} \\ 1 & v_{22} & v_{23} \\ 1 & v_{32} & v_{33} \end{bmatrix}, \quad \boldsymbol{\Lambda} = \begin{bmatrix} \lambda_0 & 0 & 0 \\ 0 & \lambda_1 & 0 \\ 0 & 0 & \lambda_2 \end{bmatrix}, \tag{36}$$

with $\lambda_0 = 0$. Note that the corresponding eigenvector of $\lambda_0 = 0$ is always a vector of identical values, here normalized as 1s. To construct the weight matrix of the models, we arrange $\boldsymbol{\omega}_i$ horizontally as

$$\boldsymbol{\Omega} = \begin{bmatrix} \omega_{11} & \omega_{12} \\ \omega_{21} & \omega_{22} \\ \omega_{31} & \omega_{32} \end{bmatrix}. \tag{37}$$

The reason for this horizontal stacking rather than column-wise is due to the graph's specification. For instance, as $\boldsymbol{V}^T$ is a $3 \times 3$ matrix, the multiplication of $\boldsymbol{\Omega}_{GF} = \boldsymbol{V}^T \boldsymbol{\Omega}$ requires the row of $\boldsymbol{\Omega}_{GF}$ to be 3. This row-wise stacking results in aggregating each column of $\boldsymbol{\Omega}_{GF}$, which is precisely what is needed: aggregating each model weight of a participant concerning the corresponding model weight of another participant. Furthermore, we introduce $\text{diag}[\zeta_1, \zeta_2, \zeta_3]$ and augment it to the filtering such that $\boldsymbol{\Psi} = H(\boldsymbol{L}; \cdot) \text{diag}[\zeta_1, \zeta_2, \zeta_3] \boldsymbol{\Omega}^{(t)}$. By defining the filter operator as $h(\lambda)$ as in equation 12, we can derive the matrix frequency coefficients of $\boldsymbol{\Omega}$ as

$$\boldsymbol{\Omega}_{GF} = h(\boldsymbol{\Lambda}) \boldsymbol{V}^T \text{diag}[\zeta_1, \ldots, \zeta_K] \boldsymbol{\Omega} = \tag{38}$$
$$\begin{bmatrix} h(\lambda_0) \sum_{j=1}^{3} \zeta_j \omega_{j1} & h(\lambda_0) \sum_{j=1}^{3} \zeta_j \omega_{j2} \\ h(\lambda_1) \sum_{j=1}^{3} v_{j2} \zeta_j \omega_{j1} & h(\lambda_1) \sum_{j=1}^{3} v_{j2} \zeta_j \omega_{j2} \\ h(\lambda_2) \sum_{j=1}^{3} v_{j3} \zeta_j \omega_{j1} & h(\lambda_2) \sum_{j=1}^{3} v_{j3} \zeta_j \omega_{j2} \end{bmatrix}.$$

Each element of the columns in $\boldsymbol{\Omega}_{GF}$ represents a frequency coefficient. Now, suppose we intend to construct the FedAvg algorithm using FEDPNP. To achieve this, we set $\beta_1 = \beta_2$ to a large value, e.g., $\beta_1 = \beta_2 = 500$. This results in $h(\lambda_0 = 0) = 1$ and $h(\lambda_1) \approx h(\lambda_2) \approx 0$. In this scenario, only the first row of $\boldsymbol{\Omega}_{GF}$ in equation 38 becomes non-zero, i.e., only DC coefficients remain. Applying the IGFT to $\boldsymbol{\Omega}_{GF}$ yields the aggregated weight matrix as

$$\boldsymbol{\Psi} = \boldsymbol{V} \boldsymbol{\Omega}_{GF} = \begin{bmatrix} \sum_{j=1}^{3} \zeta_j \omega_{j1} & \sum_{j=1}^{3} \zeta_j \omega_{j2} \\ \sum_{j=1}^{3} \zeta_j \omega_{j1} & \sum_{j=1}^{3} \zeta_j \omega_{j2} \\ \sum_{j=1}^{3} \zeta_j \omega_{j1} & \sum_{j=1}^{3} \zeta_j \omega_{j2} \end{bmatrix}. \tag{39}$$

It can be observed that each row of $\boldsymbol{\Psi}$, denoted as $\boldsymbol{\Psi}[k]$, represents a weighted average of the model weights, akin to the behavior of the FedAvg algorithm. Consequently, we have demonstrated that the proposed FEDPNP algorithm fully encompasses FedAvg. It is now straightforward to show that FedProx is also a special case of FEDPNP. Since $\boldsymbol{\Psi}[1] = \boldsymbol{\Psi}[2] = \boldsymbol{\Psi}[3] = \boldsymbol{\omega}_{\text{G}}$, the structural proximal term becomes the same proximal term used in FedProx, i.e., $\frac{\mu}{2} \|\boldsymbol{\omega}_{\text{G}} - \boldsymbol{\omega}_i\|_2^2$.

### A.6.3 ILLUSTRATION OF GRAPH FILTERS USED IN THE EXPERIMENTS

Figures 6 and 7 illustrate the graph filters defined for FEDPNP in this paper for various datasets. Note that each vertical dashed line indicates an eigenvalue of the adjacency matrix derived from Section A.5. As seen, the graph frequencies for the HARBOX dataset (i.e., the first row of Figure 6) are more dense than for those of MNIST and CIFAR-10. Furthermore, it is observed that the filter $h(\lambda; \beta_1, \beta_2)$ is a soft version of $h(\lambda; \tau)$. In addition, both filters are low-pass graph filters and are considered graph denoisers in the Fourier domain.

### A.7 FURTHER EXPERIMENTS

### A.7.1 OUT-OF-DISTRIBUTION EVALUATION

To assess FEDPNP's performance on out-of-distribution datasets, we created a new evaluation set known as the "Global Test Set." This test set comprises all local test sets from the $K$ participants, emphasizing model generalization over personalization. Ideally, a robust model should perform well on both local test sets (defined in Section 4) and the global test set.

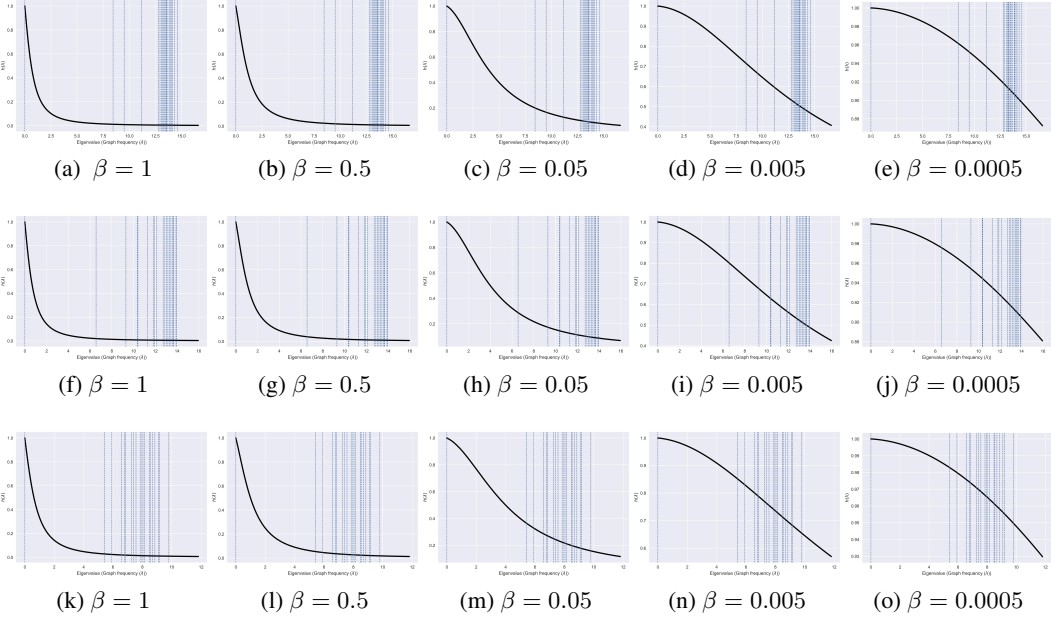

Figure 6: The graph frequency response of the designed filter $h(\lambda; \beta_1, \beta_2)$ in 12 applied on different datasets where we assume $\beta = \beta_1 = \beta_2$. The vertical lines indicate the eigenvalues of the adjacency matrix created for each dataset. The first row (a,b,c,d,e) is applied to the adjacency matrix created from HARBOX, the second row (f, g, h, i, j) from CIFAR-10, and the third row (k,l,m,n,o) from MNIST.

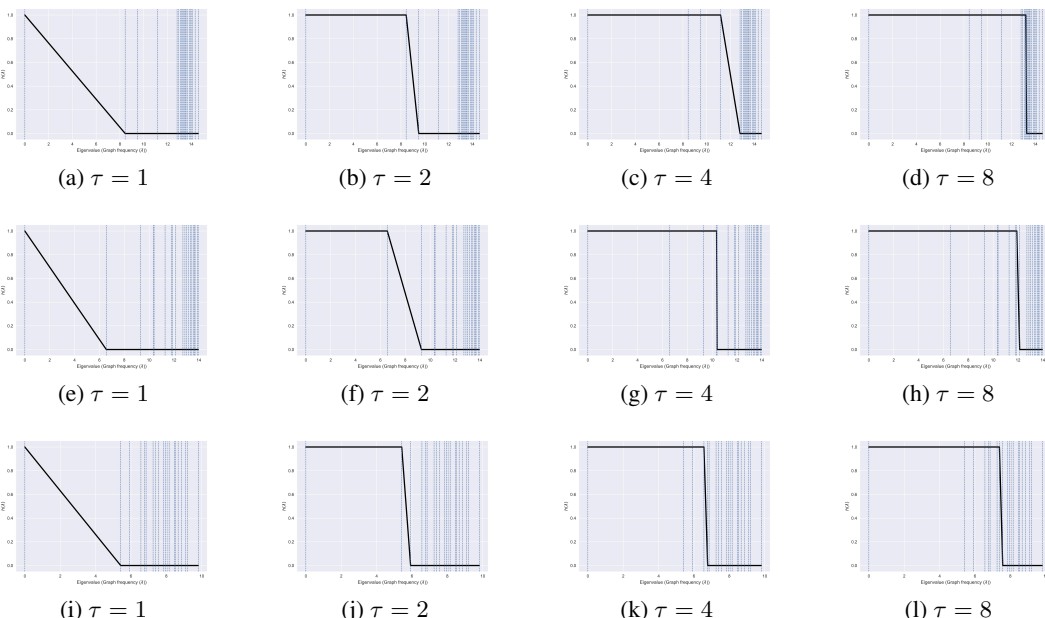

Figure 7: The graph frequency response of the designed filter $h(\lambda; \tau)$ in 16 applied to different datasets. The vertical lines indicate the eigenvalues of the adjacency matrix created for each dataset. The first row (a,b,c,d,e) is applied to the adjacency matrix created from HARBOX, the second row (f,g,h,i,j) from CIFAR-10, and the third row (k,l,m,n,o) from MNIST.

Figure 8 illustrates the convergence behavior of the FEDPNP algorithm for various $\beta = \beta_1 = \beta_2$ parameters when evaluated on the global test set. It demonstrates that as the model becomes more personalized with different $\beta$ values, test accuracy tends to decrease. However, Figure 2 reveals that personalization improves as $\beta$ is adjusted. This indicates that FEDPNP can strike a balance between model personalization and generalization using different $\beta$ values. The choice of how much personalization or generalization is required can be left to the participants.

Furthermore, fully local training (FEDPNP with $\beta = 0$) yields significantly lower performance when assessed on the global test set, as shown in Figure 8. It is also inferior to most other settings shown for the local test set in Figure 2. Therefore, performing PFL over local training is much preferable, and this preference is further underscored by the utilization of our proposed FEDPNP algorithm.

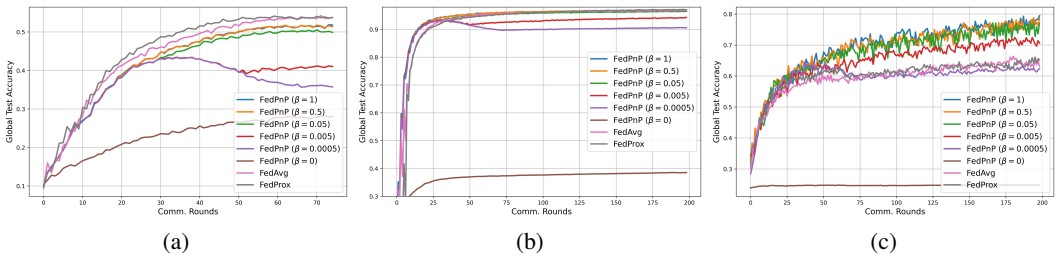

(a)           (b)           (c)

Figure 8: Effect of $\beta$ on the convergence of FEDPNP for the global test set compared to FedAvg and FedProx when $K = 20$, $\mu = 0.2$, $E = 5$. $a, d$) CIFAR-10 with $Dir(0.5)$, $b, e$) MNIST with $Dir(0.2)$, and $c, f$) HARBOX datasets. For $\beta = 0$, $\nu^{(0)} = 0$ and for the rest, it is $\nu^{(0)} = 1$

### A.7.2 THE EFFECT OF BATCH SIZE:

We investigate the impact of varying the local training batch size on FEDPNP to observe its behavior under these changes. Additionally, we determine the optimal hyperparameters for our algorithm, as analyzed in Section 4. Table 5 presents the mean accuracy of FEDPNP alongside FedAvg and FedProx for both local and global test sets.

Recall that the local test set, as discussed in Section 4, is generated using the same distribution as the local training set for each participant. In contrast, the global test set comprises the combined local test sets explained in Subsection A.7.1.

As observed, a small batch size (e.g., batch size = 16) leads to a roughly 1% decrease in performance for all algorithms. Furthermore, although FEDPNP's performance on the local test set with larger batch sizes is comparable to that of smaller ones, the global test accuracy decreases by approximately 9%.

### A.7.3 THE EFFECT OF THE NUMBER OF LOCAL UPDATES, $E$:

We also explore the influence of varying the number of local updates conducted by each participant. As shown in Table 6, optimal performance is achieved for all methods when the value of $E$ falls within the range of 4 to 8.

Table 5: Performance comparison of the proposed \textsc{FedPnP} over varying the batch-size on CIFAR-10 with $Dir(0.5)$ dataset. The results are the mean accuracy over $K = 20$ the local models and $R = 200$, $E = 5$

| batch-size | FedAvg | FedProx | FedPnP ($\mu = 0.2, \beta = 0.0005$) |
|---|---|---|---|
| | Local test Accuracy according to Section 4 | | |
| batch-size= 16 | $49.33 \pm 21.2$ | $49.12 \pm 21.3$ | $66.17 \pm 9.0$ |
| batch-size= 32 | $51.13 \pm 19.1$ | $52.59 \pm 18.6$ | $67.24 \pm 8.8$ |
| batch-size= 64 | $53.05 \pm 18.1$ | $53.29 \pm 17.8$ | $67.93 \pm 9.0$ |
| batch-size= 256 | $55.81 \pm 16.5$ | $55.61 \pm 16.6$ | $67.51 \pm 9.5$ |
| | Global test Accuracy according to Subsection A.7.1 | | |
| batch-size= 16 | $49.00 \pm 2.7$ | $49.55 \pm 2.5$ | $49.87 \pm 2.5$ |
| batch-size= 32 | $51.22 \pm 1.3$ | $52.68 \pm 1.1$ | $49.29 \pm 1.8$ |
| batch-size= 64 | $51.97 \pm 1.3$ | $53.52 \pm 1.1$ | $47.82 \pm 1.4$ |
| batch-size= 256 | $54.49 \pm 0.7$ | $55.33 \pm 0.5$ | $36.20 \pm 4.6$ |

Table 6: Performance comparison of the proposed FEDPNP over varying the number of local updates, $E$, on CIFAR-10 with $Dir(0.5)$ dataset. The results are the mean accuracy $\pm$ std over $K = 20$, the local models with a batch size of 128 and $R = 200$.

| # of local epochs $E$ | FedAvg | FedProx | FedPnP ($\mu = 0.2, \beta = 0.0005$) |
|---|---|---|---|
| | Local test Accuracy according to Section 4 | | |
| $E = 2$ | $54.05 \pm 18.1$ | $53.99 \pm 18.2$ | $66.67 \pm 9.4$ |
| $E = 4$ | $54.01 \pm 18.2$ | $54.07 \pm 18.1$ | $68.87 \pm 8.2$ |
| $E = 8$ | $53.88 \pm 20.5$ | $53.90 \pm 19.4$ | $67.20 \pm 8.8$ |
| $E = 16$ | $53.77 \pm 21.6$ | $54.09 \pm 18.8$ | $65.04 \pm 8.8$ |
| | Global test Accuracy according to Subsection A.7.1 | | |
| $E = 2$ | $54.00 \pm 1.0$ | $54.20 \pm 0.9$ | $34.67 \pm 5.1$ |
| $E = 4$ | $53.91 \pm 1.2$ | $54.32 \pm 0.9$ | $41.30 \pm 3.8$ |
| $E = 8$ | $54.05 \pm 0.9$ | $54.24 \pm 0.7$ | $47.78 \pm 1.3$ |
| $E = 16$ | $53.96 \pm 1.1$ | $54.44 \pm 0.7$ | $46.10 \pm 1.6$ |

