# OpenReview forum: "FedPnP:A Plug and Play Approach For Personalized Graph-Structured Federated Learning"
_ICLR.cc/2024/Conference — ICLR 2024 Conference Withdrawn Submission_

### Official Review · Reviewer_4efK · 2023-10-14

**Soundness:** 2 fair
**Presentation:** 3 good
**Contribution:** 2 fair
**Rating:** 5
**Confidence:** 2

**Summary:**

This paper studies the problem of personalized federated learning with a collaboration graph. The authors first construct a client-wise similarity graph with dataset similarity, and then solve the problem with an alternate optimization algorithm. First, clients optimize their local parameters with a graph-structured parameter regularization (a somewhat extension of FedProx). Then, the parameters are denoised globally to ensure a smoothness of parameters across the graph. Experiments are done with comparisons to a good range of PFL baselines, where the proposed FedPnP outperforms the baselines.

**Strengths:**

1. The paper is well organized and easy to follow. I can follow the overall organization and techniques of the paper without much efforts.

2. Theoretical analysis on FedPnP is given to show its convergence.

**Weaknesses:**

W1. It seems that the authors fail to compare the proposed FedPnP with some highly related baselines. For example, SFL (Chen et al. 2022) is a highly related baseline of FedPnP (and it also cited in this paper!) but not compared. This is a major drawback, since SFL not only involves graph structures in personalized federated learning but also learns the graph structure as well. Also, the authors could have cited an ICML 2023 paper (Ye et al. 2023). In addition, as this work aims to recover client-wise graphical relations, I think it is good to discuss its relation with federated multi-task learning (Smith et al. 2017), as the task-wise relationship can also be interpreted as a graph, and each client denotes a task in (Smith et al. 2017).

W2. Similar to W1, the authors could have made a better conceptual comparison with SFL (Chen et al. 2022). The conceptual difference and the technical innovation of FedPnP is not clear against SFL. I am not a theory guy so I may not understand this paper very well, but I believe that a conceptual comparison with SFL is good to the general audience.

W3. From table 1, it seems that FedPnP is sensitive to hyperparameter selection. For example, FedPnP with $\gamma=1$ is often very poor. In addition, the optimal $\gamma$ across datasets also vary a lot. It may be good if the authors can provide some guidelines on hyperparameter tuning.

W4. The authors may use some more realistically generated personalized FL models (e.g. those from LEAF, https://leaf.cmu.edu/). This is important especially because MNIST is simple and may not draw a very good conclusion.

(Chen et al. 2022) Personalized Federated Learning With a Graph. IJCAI 2022.

(Ye et al. 2023) Personalized Federated Learning with Inferred Collaboration Graphs, ICML 2023.

(Smiths et al. 2017) Federated Multi-Task Learning. NIPS 2017

**Questions:**

Q1, What is the conceptual difference between this work and SFL (Chen et al. 2022)? This may be a key question for me to understand the paper's contributions.

---

### Official Review · Reviewer_CRgy · 2023-10-29

**Soundness:** 2 fair
**Presentation:** 3 good
**Contribution:** 2 fair
**Rating:** 3
**Confidence:** 3

**Summary:**

This paper proposes Graph-structured FL to solve personalized FL. The authors introduce a participant relationship-based graph construction where they create such a relationship based on dataset similarity.

**Strengths:**

It is good see that the authors exploit graph representation within the scope of FL.
Capturing the interconnections among participants is interesting.

**Weaknesses:**

- Constructing the graph, edge weights is too complex. The complexity is O(N^2) which is impractical.
- Adjacency matrix is solely based on each data sample which makes this application too slow.

**Questions:**

- What happens when there is a new participant ?
I suspect, when there is a new participant, Adj matrix needs to be reconstructred from scract and this should be avoidable.

---

### Official Review · Reviewer_BmWz · 2023-10-30

**Soundness:** 1 poor
**Presentation:** 1 poor
**Contribution:** 1 poor
**Rating:** 3
**Confidence:** 4

**Summary:**

Authors formulate personalized federated learning as a graph-based optimization problem linked to inverse problems like compressed sensing. This formulation involves a known graph structure that reflects similarities between local models. The main contribution is a novel optimization method (algorithm 1) for solving the resulting optimization problem.

**Strengths:**

I appreciate the study of advanced optimization (Half-Quadratic- Splitting (HQS) technique) techniques for optimization problems arising in personalized federated learning

**Weaknesses:**

The signficance/novely of the work needs to be discussed more convincingly. What does your method offer compared to existing work on total variation minimization for federated learning. A seminal work here is

David Hallac, Jure Leskovec, and Stephen Boyd. 2015. Network Lasso: Clustering and Optimization in Large Graphs. In Proceedings of the 21th ACM SIGKDD International Conference on Knowledge Discovery and Data Mining (KDD '15). Association for Computing Machinery, New York, NY, USA, 387–396. https://doi.org/10.1145/2783258.2783313

There is a substantial body of work on the computational properties (convergence speed) of distribution optimization methods for network Lasso and more general total variation minimization methods, see, e.g. [Ref2] and references therein. The analysis of TV minimization methods in [Ref2] also allows to characterize the clustering of learned local models. This clustering of local models is one approach to personalized federated learning (see [Ref3]). Would a similar analysis of computational and clustering properties also be possible for your Algorithm 1 ?


[Ref2] Y. SarcheshmehPour, Y. Tian, L. Zhang and A. Jung, "Clustered Federated Learning via Generalized Total Variation Minimization," in IEEE Transactions on Signal Processing, doi: 10.1109/TSP.2023.3322848.

[Ref3] Werner, M., He, L., Praneeth Karimireddy, S., Jordan, M., and Jaggi, M., “Provably Personalized and Robust Federated Learning”, <i>arXiv e-prints</i>, 2023. doi:10.48550/arXiv.2306.08393.

* it seems that some numbered equations are not referred to/discussed at all.

* pls avoid jargon such as "intriguing proximal term" or "a perfect setting for our federated experiments"

* the font size in the figures (e.g., for axis and ticks labels) is way too small.

**Questions:**

* What is the precise relation between different optimization problems P1 .. P4 ?

* Is there some clustering assumption behing P1 .. P4, i.e., will the solutions be piece-wise constant model parameters ?

* What is practical relevance of the convergence analysis summarized in Thm 1,2 and Cor. 1.  In particular, are the required assumptions satisfied for important application domains and the settings of the numerical experiments?

* How challenging is the construction of a useful graph/adjacency matrix A (e.g., in the numerical experiments)?

---

### Official Review · Reviewer_vGbJ · 2023-10-31

**Soundness:** 3 good
**Presentation:** 2 fair
**Contribution:** 3 good
**Rating:** 5
**Confidence:** 3

**Summary:**

Authors generalize the personalized federated learning algorithm, FedProx, for graph-structured federated learning settings.

**Strengths:**

(1) Extends FedProx for graph-structured FL setting.

(2) The relationship between inverse problems and graph-structured FL is interesting.

(3) Theoretical convergence provided.

**Weaknesses:**

(1) The problem is over graph-structured federated learning. All the datasets used in the experiment have no explicit relationship at all. I positively support using graph construction methods, which is a valid method. However, It is better to use traffic/weather forecasting datasets such as METR-LA & PEMS-BAY [1] so that reviewers can see that the proposed method can utilize the actual underlying relationships. This is why the datasets used in this paper do not simulate the real-world graph PFL problems.

(2) There are some personalized FL and graph-structured FL methods, such as CNFGNN [2] & APFL[3] you can compare against it.


[1] Li, Yaguang, et al. "Diffusion convolutional recurrent neural network: Data-driven traffic forecasting." arXiv preprint arXiv:1707.01926 (2017).
[2] Chuizheng Meng, Sirisha Rambhatla, and Yan Liu. 2021. Cross-Node Federated Graph Neural Network for Spatio-Temporal Data Modeling. In Proceedings of the 27th ACM SIGKDD Conference on Knowledge Discovery & Data Mining (KDD '21). Association for Computing Machinery, New York, NY, USA, 1202–1211. https://doi.org/10.1145/3447548.3467371
[3]Deng, Yuyang, Mohammad Mahdi Kamani, and Mehrdad Mahdavi. "Adaptive personalized federated learning." arXiv preprint arXiv:2003.13461 (2020).

**Questions:**

(1) in P1, why not optimize the trace of L_smooth and L_linear? The problem would be in one-dimensional space.